# Structural insights into the regulation, ligand recognition, and oligomerization of bacterial STING

Mei-Hui Hou[1,7], Yu-Chuan Wang [1,7], Chia-Shin Yang[1], Kuei-Fen Liao[2], Je-Wei Chang [2], Orion Shih[2], Yi-Qi Yeh [2], Manoj Kumar Sriramoju[3], Tzu-Wen Weng[3,4], U-Ser Jeng[2,5], Shang-Te Danny Hsu [3,4] & Yeh Chen [6] ✉

The cyclic GMP-AMP synthase (cGAS)/stimulator of interferon gene (STING) signaling pathway plays a critical protective role against viral infections. Metazoan STING undergoes multilayers of regulation to ensure specific signal transduction. However, the mechanisms underlying the regulation of bacterial STING remain unclear. In this study, we determined the crystal structure of anti-parallel dimeric form of bacterial STING, which keeps itself in an inactive state by preventing cyclic dinucleotides access. Conformational transition between inactive and active states of bacterial STINGs provides an on-off switch for downstream signaling. Some bacterial STINGs living in extreme environment contain an insertion sequence, which we show codes for an additional long lid that covers the ligand-binding pocket. This lid helps regulate anti-phage activities. Furthermore, bacterial STING can bind cyclic di-AMP in a triangle-shaped conformation via a more compact ligand-binding pocket, forming spiral-shaped protofibrils and higher-order fibril filaments. Based on the differences between cyclic-dinucleotide recognition, oligomerization, and downstream activation of different bacterial STINGs, we proposed a model to explain structure-function evolution of bacterial STINGs.

The mammalian cyclic GMP-AMP synthase (cGAS)/stimulator of interferon gene (STING) pathway plays an important role in innate immunity against invading viral and bacterial pathogens[1–3]. In the absence of an activating signal, metazoan STING is auto-inhibited via binding to its own C-terminal tail (CTT), which blocks STING oligomerization[4]. In the initial stage of infection, the presence of double-stranded DNA from viruses or bacteria in the cytosol activates mammalian cGAS, which then generates a second messenger, 2'3'-cyclic GMP-AMP (cGA). Binding of 2'3'-cGA to STING releases its CTT from the oligomerization interface, causing lid closure and STING oligomerization, which is further stabilized through Cys148-mediated disulfide-linkage[4]. The activated STING polymer acts as a platform for the recruitment of TANK-binding kinase 1 (TBK1), which then phosphorylates interferon regulatory factor 3 (IRF3) to induce type I interferon production[1,5]. Additionally, the cGAS-STING pathway activates the transcription factor nuclear factor-κB (NF-κB), leading to the expression of proinflammatory cytokines, such as tumor necrosis factor-α and interleukin 6[6]. In the late stage of infection, deactivation of STING signaling is mediated by lysosome degradation[7]. Sustained activation of STING can be achieved by removing its K48-linked polyubiquitin chains[8]. Furthermore, palmitoylation contributes to STING activation in Golgi[9]. Finally,

[1]Genomics BioSci. & Tech. Co. Ltd., New Taipei 221411, Taiwan. [2]National Synchrotron Radiation Research Center, Hsinchu Science Park, Hsinchu 300092, Taiwan. [3]Institute of Biological Chemistry, Academia Sinica, Taipei 115024, Taiwan. [4]Institute of Biochemical Sciences, National Taiwan University, Taipei 106319, Taiwan. [5]Department of Chemical Engineering & College of Semiconductor Research, National Tsing Hua University, Hsinchu 300044, Taiwan. [6]Department of Food Science and Biotechnology, National Chung Hsing University, Taichung 402202, Taiwan. [7]These authors contributed equally: Mei-Hui Hou, Yu-Chuan Wang. ✉e-mail: chyeah6599@nchu.edu.tw

negative regulation by PPM1A phosphatase antagonizes STING phosphorylation through TBK1, completing the regulatory circuit of cGAS-STING signaling[10]. Further, regulatory abnormalities in the human cGAS-STING pathway have been associated with various human diseases[11]. A genetic disorder known as STING-associated vasculopathy with onset in infancy can cause systemic inflammation and interstitial lung disease[12]. Neurodegenerative diseases such as Parkinson's disease[13] and autoimmune diseases such as Aicardi-Goutières syndrome[14] are also associated with the cGAS-STING pathway.

Phylogenetic analysis revealed that the cGAS-STING pathway is evolutionarily conserved in mammals, birds, reptiles, amphibians, fish, insects, mollusks, cnidarians, and the unicellular eukaryote, choanoflagellate[15]. In contrast to vertebrate STINGs, early metazoan STINGs lack CTT and likely rely on autophagy for the clearance of invading pathogens[16]. A recent groundbreaking study discovered the ancient origins of the cGAS-STING pathway in bacteria, which play a similar role in the immune system to combat bacteriophage infections[17]. During phage infection, the bacterial cGAS counterpart, cGAS/DncV-like nucleotidyltransferase, can produce 3′3′-cyclic di-GMP (cGG) to activate bacterial STING[17]. Binding of cGG to bacterial STING causes it to oligomerize into long filaments and activates the nicotinamide adenine dinucleotide (NAD)$^+$ cleavage activity of its fused N-terminal Toll/interleukin-1 receptor (TIR) domain. Depletion of the electron carrier NAD$^+$ eventually leads to bacterial growth arrest or cell death, restricting the replication of invading phages in infected bacteria, a strategy called "abortive infection." As a result, the precise regulation of bacterial STING signaling in the presence or absence of invading phages becomes essential for bacterial survival. A ubiquitin-transfer-like mechanism has been shown to regulate the enzyme activity of bacterial cGAS[18]. However, the mechanisms of bacterial STING regulation remain unclear.

Ligand recognition is crucial for ensuring specific STING signaling[19]. Human STING binds endogenous 2′3′-cGA produced via cGAS during viral infections[3]. In addition, human STING can recognize bacterial cyclic dinucleotides (CDNs), including 3′3′-cGG, 3′3′-cGA, and 3′3′-cyclic di-AMP (cAA), as pathogen-associated molecular patterns to elicit both IFN-I responses and NF-κB signaling[20]. Notably, it was reported that various STING variants bind CDNs differently. Human STING$^{R232}$ responds to all CDNs, whereas the human STING$^{H232}$ variant preferentially recognizes 2′3′-cGA over others[2]. In prokaryotes, STING can be divided into two classes based on the conserved sequence motif Rx(Y/F) and RxR in the STING β-strand lid, which is responsible for cGG and cGA recognition[21]. A recent study further identified dozens of bacterial STING fused with versatile effector domains; some of these bacterial STINGs did not contain conserved motifs, indicating their unknown ligand specificity[22]. In addition, in some bacterial species, the cAA-synthesizing enzyme DNA integrity scanning protein (DisA) is directly linked to the gene encoding STING[22], suggesting the existence of cAA-mediated STING signaling. cAA plays vital roles in bacterial survival, including cell wall homeostasis[23], DNA damage sensing[24], osmoregulation[25], and pathogenesis;[26] therefore, it is of great interest to understand whether STING recognizes the second messenger cAA and how cAA activates STING signaling for immune defense against phage infections.

In this study, we demonstrated two distinct mechanisms underlying the regulation of bacterial STINGs and a structural basis for cAA recognition and oligomerization. We summarized the current understanding and proposed a model to explain the ligand preference, oligomerization, and evolution of bacterial STINGs under selective pressure. In conclusion, this study provides key insights into the bacterial cGAS-STING signaling pathway, thereby expanding our understanding of the innate immunity of prokaryotes.

## Results

### Bacterial STING adopts an anti-parallel dimeric conformation

Despite lacking CTT, it is hypothesized that bacterial STINGs utilize unknown structural element to inhibit itself in the absence of activating second messenger. To investigate the potential mechanisms underlying STING inhibition, we set to structurally characterize bacterial STING in the absence of ligand and compare with its ligand-bound form. The crystal structures of bacterial STING from *Riemerella anatipestifer* (*Ri*STING) in its ligand-free and cGG-bound form were determined to 2.1-Å and 1.46-Å resolution, respectively (Supplementary Table 1). Ligand-free *Ri*STING assembles into a distinct dimeric architecture, significantly differing from the V-shaped architecture of cGG-bound *Ri*STING (Fig. 1a, b and Supplementary Fig. 1a) and previously known ligand-bound STINGs[17,21,27,28] (Supplementary Fig. 2a–f). The major α-helices of one ligand-free *Ri*STING protomer, α1, α3, and α4, have "anti-parallel" orientations with α1′, α3′, and α4′ helices of the other protomer, respectively, henceforth named "anti-parallel dimer" (Fig. 1c). The major driving force for the V-shaped dimerization is formation of a hydrophobic core constituting the four-helix bundle (α1/α3/α1′/α3′); in contrast, the anti-parallel dimerization makes more extensive contact with each other by forming six-helix bundles (α4/α1/α3/α1′/α3′/α4′) (Fig. 1a, b).

Recognition of cGG by *Ri*STING showed significant differences from previously determined cGG-bound STING[21]. Despite the conserved D252 interacting with the N2 atom of the guanine base as previously reported[17,21], the bound cGG is sandwiched between the guanidinium group of R235 and aromatic ring of F171 (hereafter "sandwich binding"), in contrast to the four-layer stack interaction with cGG by *Prevotella corporis* STING (*Pc*STING)[21] (Fig. 1d, e). Another conserved arginine residue R233 of *Ri*STING deviated from the bound cGG and only forms one H-bond with the guanine base in contrast to the R233 of *Pc*STING, which forms closed hydrogen bonding interactions with the guanine base (Fig. 1f). This provided a possible explanation for differential binding affinities to cGG by different bacterial STINGs. Superimposition of the protomers between anti-parallel *Ri*STING with cGG-bound *Ri*STING demonstrated that the binding of cGG induced large conformational changes in the β-strand lid, α2, and α3 helix (Supplementary Fig. 2g). Alignment of the anti-parallel *Ri*STING dimer with the cGG-bound V-shaped *Ri*STING dimer revealed a rigid-body rotation of a whole protomer (~70°) plus a dramatic conformational change in the α3 helix (~120°) (Supplementary Fig. 2h). The anti-parallel dimerization is mediated by α3-α3′, α3-α4 loop-α1′, and α1-α3′α4′ loop interactions, forming an extensive interface of approximately 1400 Å² (Supplementary Fig. 3a). The anti-parallel α3-α3′ interaction is mediated by hydrophobic contacts between residues T255, V256, S258, G259, and Y262 and one hydrogen bond (S258/S258′) (Supplementary Fig. 3b). The second part of dimerization is mediated by residues L267, L268, N270, F272, N273, Y279, I282, L283, and E286 from the α3-α4 loop of one protomer and residues P159, T162, A165, V166, E169, N170, and P174 from the α1 helix, along with residues F228, Y230, and R235 from the β-strand lid of the other protomer (Supplementary Fig. 3c, d). The anti-parallel dimerization of *Ri*STING results in the sealing of half-site of the ligand-binding pocket of one protomer via interaction with the α3-α4 loop of the other protomer, preventing ligand access to binding pocket. Superimposition of anti-parallel *Ri*STING with cGG-bound *Ri*STING revealed that the bound cGG will cause severe clashes with the side-chains of Y262, N266, and L267 and the main-chain backbone of residues Y262-P269 of anti-parallel *Ri*STING (Supplementary Fig. 2i). In the absence of a ligand, the conserved arginine residues R233 and R235 of anti-parallel *Ri*STING moved away from the ligand-binding pocket, accompanied by the extension of the β-strand, further hindering the recognition of CDNs (Supplementary Fig. 2j). As a result, the anti-parallel dimeric architecture of ligand-free *Ri*STING is considered as an inactive state of bacterial STING owing to deformation of the ligand-binding pocket at the dimer interface.

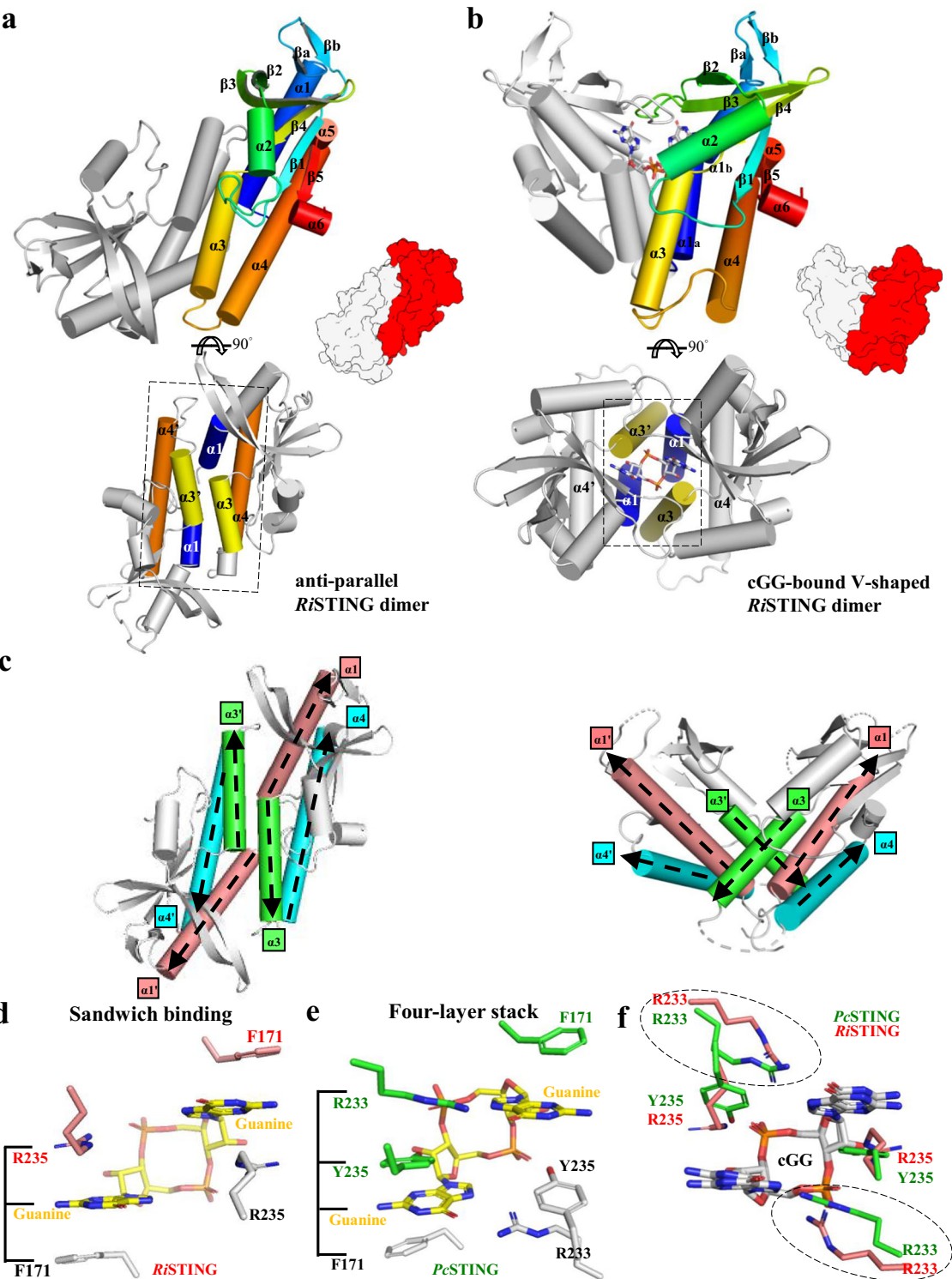

**Fig. 1 | Crystal structures of anti-parallel *Ri*STING dimer and cGG-bound V-shaped *Ri*STING dimer. a, b** Top, cartoon presentation of (**a**) anti-parallel *Ri*STING dimer and (**b**) cGG-bound V-shaped *Ri*STING dimer with five(six) α-helices and five β-strands indicated. The simplified protein model demonstrating the rigid-body rotation of one of the two protomers between these two conformations were depicted in bottom right. Bottom, the top views of (**a**) anti-parallel *Ri*STING dimer and (**b**) cGG-bound V-shaped *Ri*STING dimer. The α helices participates in dimerization interfaces are colored accordingly. The hydrophobic core of V-shaped *Ri*STING dimer constitutes a four-helix bundle (α1/α3/α1′/α3′) in contrast to the six-helix bundle (α4/α1/α3/α3′/α1′/α4′) of the anti-parallel *Ri*STING dimer. **c** Structural comparison of two ligand-free bacterial STINGs. Left, anti-parallel *Ri*STING dimer.

Right, V-shaped *Cg*STING dimer (PDB: 6WT5[17]). The major α-helices, α1, α3, and α4 are colored in salmon, green, and cyan, respectively. The N-to-C direction of the major α-helices were indicated by black dashed arrows. **d** Both the guanine bases of the bound cGG (yellow) were sandwiched by the sidechain of F171 and R235 of *Ri*STING (salmon and white). **e** The typical four-layer stack interaction (F171/Gua/Y235/R233) with cGG by *Pc*STING (green and white, PDB: 7EBD). The interacting residues and cGG are shown in sticks. **f** Structural comparison of the cGG-binding residues in *Ri*STING_cGG (salmon) and *Pc*STING_cGG (green). The residues and the bound cGG (white) are shown in sticks. The difference between R233 of *Ri*STING and *Pc*STING are indicated by black dashed circles.

Based on the above crystallographic observations and current understandings, it is suggested that bacterial STING can form three different conformation states: (1) inactive anti-parallel ligand-free form, (2) inactive V-shaped ligand-free form, and (3) active V-shaped ligand-bound form. In the absence of cGG, bacterial STINGs form both anti-parallel and V-shaped dimeric architectures, which exist in a dynamic equilibrium. The anti-parallel dimerization of bacterial STING prevents ligand access to the binding pocket and thereby keep itself consistently in an inactive state, whereas the V-shaped dimer is ready to bind cognate ligand (Fig. 2a). To validate this hypothesis, isothermal titration calorimetry (ITC), dynamic light scattering (DLS), and small- and wide-angle X-ray scattering (SWAXS) were performed. First, ITC confirmed the direct binding of cGG to *Ri*STING with nanomolar affinity (Fig. 2b). DLS experiments were subsequently performed to measure the conformational change in *Ri*STING induced by ligand binding in solution. In the absence of cGG, the major particle size (67%) of *Ri*STING was measured to be 9.0 nm (Fig. 2c). The addition of two molar excesses of cGG changed the major particle size (84%) of *Ri*STING to 40.6 nm (Fig. 2c). Over fourfold increase in the particle size of *Ri*STING provided solid evidence that cGG induced large conformational changes of *Ri*STING. To further access the solution structure of *Ri*STING in the absence or presence of cGG, SWAXS experiments were performed. The results showed that apo *Ri*STING in solution exhibits a slightly larger $R_g$ value of 22.7 ± 0.1 Å than that of cGG-bound *Ri*STING dimer (22.3 ± 0.1 Å, Supplementary Table 2). The SWAXS data of the cGG-bound *Ri*STING dimer in solution are in good agreement with the SWAXS profile calculated directly from the corresponding crystal structure in this study (PDB 8HY9, $\chi^2 = 1.98$), indicating that the solution structure is of similar structural features as that in the crystalline form (Fig. 2d). Most of the SWAXS data of the apo *Ri*STING dimer can be described reasonably well by the corresponding crystalline form in this study (PDB 8HYN, $\chi^2 = 8.57$); however, the smeared SWAXS data in the *q*-range data near 0.3 Å$^{-1}$ deviate from the calculated SWAXS profile (namely diminished scattering hump from that of the crystalline form), suggesting a less rigid solution structure of apo *Ri*STING dimer compared to the crystalline form (Fig. 2d). Moreover, in silico reconstruction of GASBOR models from the SWAXS data revealed that the ab initio shapes of both apo *Ri*STING dimer and cGG-bound *Ri*STING dimer can consistently dock well with the corresponding crystal structures, further supporting the consistency between them (Fig. 2e, f). Overall, these data indicate a transition of the solution structure of apo *Ri*STING dimer to a more rigid form after cGG binding, supporting the proposed hypothesis.

### Bacterial STING from extremophiles contains an additional long β-strand lid

Through analysis of hundreds of bacterial STINGs, we found that some bacterial STINGs thriving in extreme environment have insertion sequence at the beginning of C-terminal STING domain (Fig. 3a and Supplementary Table 3). We determined the crystal structure of one of the longer bacterial STINGs, *Larkinella arboricola* STING (*Lr*STING) to 2.73 Å resolution (Supplementary Table 4); the asymmetric unit contains six similar polypeptide chains of *Lr*STING (root mean square deviation of 0.3–0.5 Å for 127–149 Cα pairs), which assemble onto three V-shaped dimers with a cGG bound in the central ligand-binding pocket (Supplementary Fig. 1b–d, 4a). *Lr*STING dimers are similar to previously determined cGG-bound bacterial STINGs (r.m.s.d. of 1.3–1.9 Å for 185–195 Cα pairs), with the exception that *Lr*STING contains strikingly longer β strands a and b (βa and βb, Fig. 3b, c and Supplementary Fig. 4b). Notably, these structural features are absent in all solved eukaryotic STING structures (Fig. 3d). The longer βa and βb form a continuous β sheet with β1 and β2 and make extensive interactions with βa and βb of the other protomer, forming an additional long lid above the original lid (Fig. 3e). Superimposition of the three *Lr*STING dimers in the asymmetric unit revealed that their overall structures are nearly identical, except for significant conformational changes of the additional long lid, suggesting higher structural flexibility as indicated by B-factor values (Fig. 3f and Supplementary Fig. 4a). Closure of the long lid of *Lr*STING (chains AB) results in an increase in the dimerization interface by ~1000 Å$^2$ more than previously determined bacterial STINGs[21]. This interface contains two interchain hydrogen-bonding pairs (K202/T206) and numerous van der Waal interactions between Q192, T200, and P203 (Fig. 3e). In contrast, the long lid of *Lr*STING (chains CD) form one interchain H-bond (K202/T206) and one salt-bridge (R199/D208) at only one end, whereas the long lid of *Lr*STING (chains EF) does not interact with each other (Fig. 3g, h). It is thus suggested that the three different lid conformations of *Lr*STING are probably snapshots of a dynamic process of lid opening and closing, suggests a regulatory role to control CDN access to the ligand-binding pocket. To prove this hypothesis, we replaced residues 190−210 with Gly and Ser to create a lid-to-loop mutant, *Lr*TIR-STING[lid→loop]. In cell toxicity assay, *E. coli* cells expressing wild-type *Lr*TIR-STING slightly inhibit the cell growth by OD$_{600}$ of 0.1–0.15 compared to control cells with empty vector (Fig. 3i). By contrast, *E. coli* cells expressing *Lr*TIR-STING[lid→loop] mutant experienced complete growth arrest, suggesting that the NADase activity of *Lr*TIR-STING was strongly enhanced in the absence of the additional long lid (Fig. 3i). The phage infection assay further demonstrated that overexpression of *Lr*TIR-STING[lid→loop] can help bacteria resist phage invasion rather than wild-type *Lr*STING (Fig. 3j), consistent with the results of cell toxicity. In summary, these data suggested that the formation of an additional long lid of *Lr*STING results in autoinhibition, leading to attenuated *E. coli* toxicity and anti-phage ability (Fig. 3k). Mutating the lid to random coil relieve the autoinhibition, thereby elevating the cell toxicity and anti-phage ability of *Lr*STING to a level similar to canonical bacterial STINGs (Fig. 3k).

### Bacterial STING recognizes cAA in a triangle-shaped conformation

To understand whether bacterial STINGs bind the second messenger cAA and mediates antiphage functions, a dozen of candidates from different species were screened. *Epilithonimonas lactis* STING (*El*STING) was identified to bind cAA with micromolar affinity (Supplementary Fig. 5). However, cAA could not activate the NAD+ cleavage activity of full-length *El*TIR-STING even in the presence of high concentration of cAA (100 μM) (Fig. 4a). *E. coli* toxicity assay showed no differences between *E. coli* cells expressing full-length *El*TIR-STING with and without addition of cAA and control cells with empty vector (Fig. 4b), consistent with the above enzymatic assay. Furthermore, *E. coli* cells expressing full-length *El*TIR-STING did not confer resistance to phage infection with or without addition of 100 μM cAA into growth medium or co-expression of the diadenylate cyclase CdaS[L44F] from *Bacillus subtilis*[29] (Fig. 4c, d). In conclusion, cAA binds to *El*TIR-STING but has no effect on its NAD+ cleavage activity and anti-phage ability.

To reveal the molecular mechanism underlying cAA recognition by *El*STING, we determined the 2.57 Å and 2.55 Å crystal structures of ligand-free and cAA-bound form of *El*STING, respectively, which forms canonical V-shaped dimer conformations (Fig. 4e, f and Supplementary Table 5). The omitted electron densities of bound cAA molecules are shown inside the ligand-binding pocket of *El*STING dimers (Supplementary Fig. 1e, f). Binding of cAA stabilizes the formation of the β-strand lid and further induces complete lid closure to seal the ligand-binding pocket of *El*STING (Supplementary Fig. 6a). This results in formation of a more compact dimeric architecture than cGG-bound bacterial STINGs (Supplementary Fig. 6b). Recognition of cAA by *El*STING is mainly through shape complementarity. In particular, the hydrophobic sidechain of Y165, F169, and L234, a lid residue P233 along with phosphate-backbone-interacting residues N168, T254, and T255 together constitute a more compact and limited ligand-binding pocket to clamp cAA inside (Fig. 4g). In contrast to the cGG binding

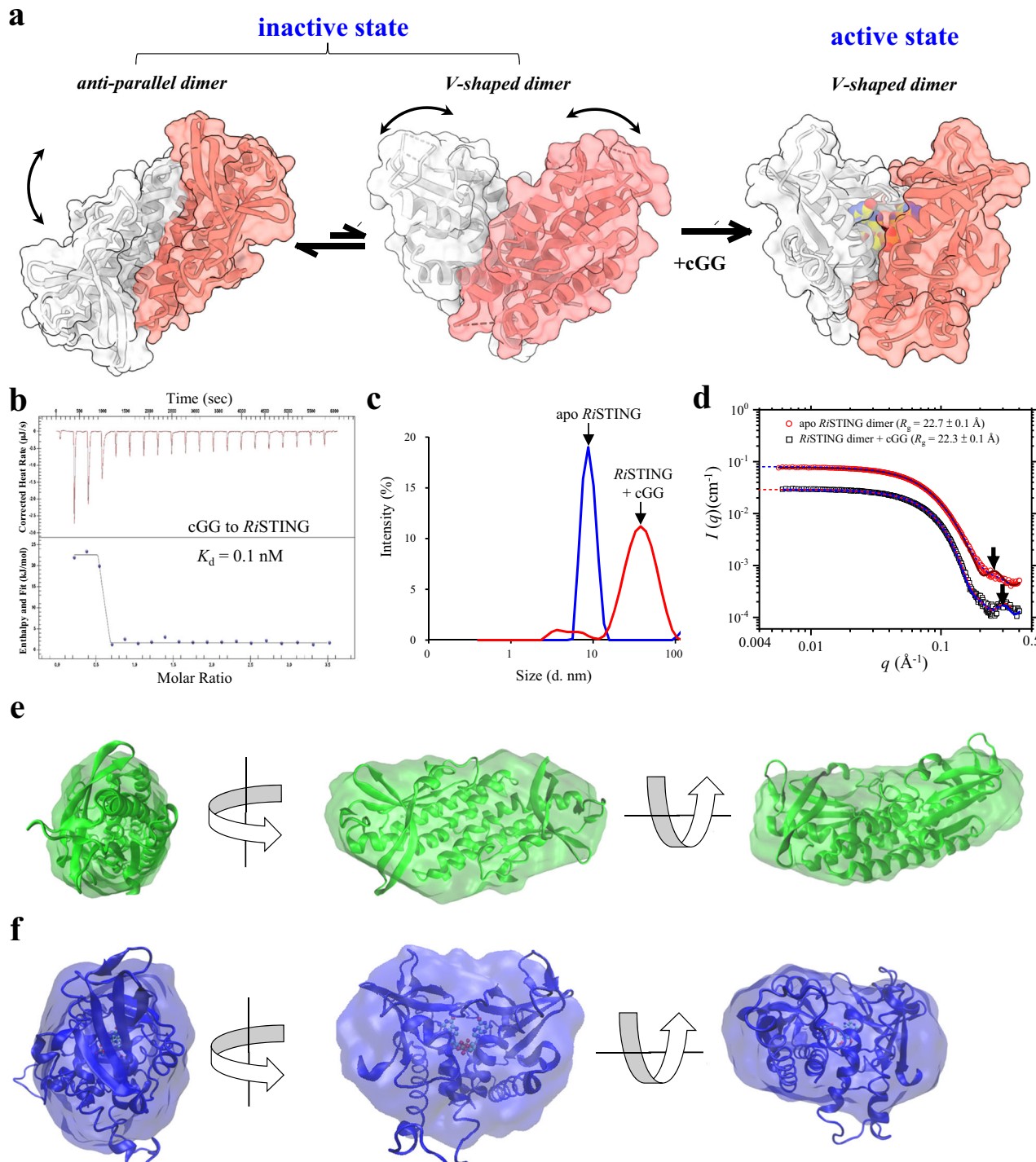

**Fig. 2 | Conformational transition between inactive and active states of bacterial STINGs. a** The cartoon model explaining the conformational changes between the three different conformational states of bacterial STINGs. In the absence of ligand, bacterial STINGs exist in equilibrium between an anti-parallel dimer and a V-shaped dimer. Upon phage infections, the produced second messenger CDN shift the equilibrium toward V-shaped ligand-bound form by stabilizing the formation of V-shaped dimers, which can further oligomerize into long STING filaments. **b** ITC analysis of cGG binding to *Ri*STING. The dissociation constant was determined to be 0.1 nM. **c** Dynamic light scattering analysis of *Ri*STING in the absence (blue line) or presence (red line) of two molar excess of cGG. The experiments in (**b**) and (**c**) were repeated at three times and the representative of them are shown. **d** CRYSOL comparison between the SWAXS data of the solution structures (apo *Ri*STING dimer, red circle; cGG-bound *Ri*STING dimer, black rectangle) and the calculated SWAXS profiles generated by their corresponding crystal structures (solid curves). Also shown are the fitting profiles (dotted curves) using **e, f** the best-fitted GASBOR models shown in shaded areas docked with their corresponding crystal structures of (**e**) apo *Ri*STING (PDB 8HYN) and (**f**) *Ri*STING_cGG complex (PDB 8HY9).

mode, there is no room leaving for cation-π or π-π interactions between the sidechain of R230/F232 and the bound cAA (Fig. 4g, h). R230 and F232 of each *El*STING protomer are projected out of the ligand-binding pocket and form different conformations in contrast to the nearly identical conformation of R233 and Y235 of *Pc*STING, which directly interact with cGG (Supplementary Fig. 6c, d). This flexibility could be partially attributed to the formation of relatively flexible β-strand lids, as indicated by B-factor values (Supplementary Fig. 6e, f).

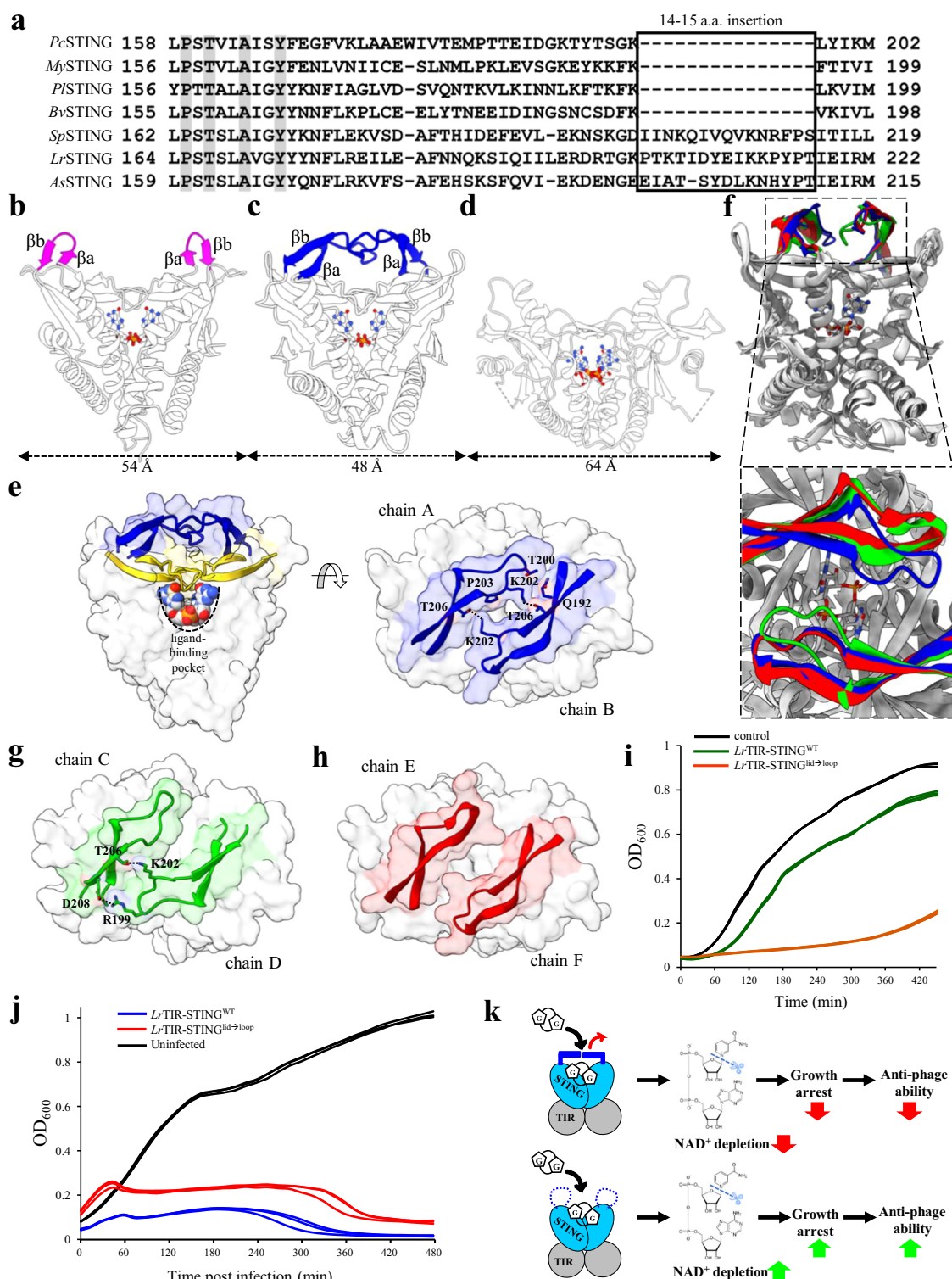

As a result, the bound cAA formed a self-stacked, triangle-shaped conformation in which the two adenine bases came close to each other by inward rotation of 26° and shortened the distance between the 6-NH₂ group of the adenine bases to 3.3 Å in comparison with 5.4 – 6.7 Å of the U-type cAA bound in eukaryotic STINGs (Fig. 4i). Here, we named this triangle-shaped cAA conformation as Δ-type to distinguish it from the four previously identified cAA conformations, including U-type, V-type, E-type, and O-type[30]. The more closely packed cAA conformation may compensates for lack of specific bonding with the protein sidechain and leads to different

oligomerization mechanism and downstream activation. The base recognition of cAA by *El*STING was mediated via indirect hydrogen bonding through water molecules: w1 and w2 bridges the side-chains of the D251 and N3 atoms of the adenine base, whereas w3 mediates the hydrogen bonding between the main-chain of N168 and the N7 atom of the adenine base (Supplementary Fig. 6g). The 3′3′-phosphodiester bond linkage of the bound cAA is symmetrically recognized by conserved threonine residues among bacterial STINGs (Supplementary Fig. 6g). Based on structural comparison and sequence alignment, we identified two key residues, T172 and A176, which are responsible

**Fig. 3 | The crystal structure of *Lr*STING in complex with cGG. a** Sequence alignment of canonical bacterial STING with *Lr*STING-like proteins. An insertion sequence of 14-15 amino acids among *Lr*STING-like proteins has been highlighted. *Pl*STING: *Pedobacter luteus* STING; *Bv*STING: *Balneola vulgaris* STING; *Sp*STING: *Spirosoma sp.* STING; *As*STING: *Arthrobacter sp.* STING. **b–d** Structural comparison of (**b**) *My*STING_cGG (PDB: 7EBL), (**c**) *Lr*STING_cGG and (**d**) human STING_2'3'-cGA (PDB: 4KSY[55]). The CDNs are shown in ball-and-stick model. The β-strand a and b of *My*STING (magenta) and *Lr*STING (blue) are highlighted for comparison. The dimension of them (Å) is indicated. **e** Left, surface representation of *Lr*STING (chains AB), which contains an additional long β-strand lid (blue) above the original β-strand lid (yellow), covering the ligand-binding pocket (black dashed lines). Right, the top-down view of *Lr*STING long lid. The residues involved in dimerization are shown in sticks. **f** Top, the ribbon diagram of three superimposed *Lr*STING dimers in the same asymmetric unit cell. The additional long lids of them are highlighted and colored (blue for chain AB; green for chain CD; red for chain EF).

Bottom, the top view of the three superimposed *Lr*STING dimers. **g, h** the top-down view of *Lr*STING long lid in (**g**) chain CD and (**h**) chain EF. The additional long lids are shown in surface and the key residues involved in dimerization are shown in sticks. The H-bonds or salt-bridges are indicated by black dashed lines. **i** Growth curves of *E. coli* cells overexpressing wild-type (green lines) and lid-to-loop mutant (orange lines) of full-length *Lr*STING protein compared with control (empty vector, black lines). **j** Growth curves of *E. coli* cells with and without phage T2 infection at a MOI of 0.01. Induction of the expression of lid-to-loop mutant (red lines) of full-length *Lr*STING confer significant anti-phage ability than the wild-type (blue lines). The *E. coli* cells without IPTG induction and T2 phage infection (black lines) serve as negative control. The experiments in (**i**) and (**j**) were performed for *n* = 3 biological replicates and each of them is shown. **k** Schematic models demonstrating the ligand binding and downstream signaling of bacterial STINGs with (top) or without (bottom) additional lid.

for the tight clamping of cAA by *El*STING. The residue T172 of *El*STING was located near the kink of the α1 helix, and its side-chain could form an additional H-bond with the main-chain of nearby residue N168, reinforcing the ligand-bound, closed state (Supplementary Fig. 6h). In *Pc*STING, the binding of cGG to the ligand-binding site induces conformational changes in the α1 and α4 helices, which results in lid closure and oligomerization interface exposure[21]. The W178 in the α1 helix of *Pc*STING forms extensive hydrophobic interactions with nearby residues and is an intermediate residue conveying the conformational change to the oligomerization interface (Fig. 4j). In contrast, W178 is replaced by an alanine residue (A176) in *El*STING, which may have conferred structural flexibility to *El*STING to form a more tightly clamped conformation to recognize cAA (Fig. 4j). Altogether, we here demonstrate the molecular basis for cAA recognition by bacterial STING through a more compact and limited ligand-binding pocket.

## Bacterial STINGs oligomerize into long filaments with different conformations

Binding of cAA induced lid closure of *El*STING, which further oligomerizes into spiral-shaped filaments via lateral dimer-dimer interactions with a rotation of 30° per dimer along the direction of filament extension (Fig. 5a). By contrast, cGG-bound *Ri*STING formed a linear filament mediated via lateral dimer-dimer interactions without rotation (Fig. 5b) like previously reported linear STING filaments[21,31]. The buried surface area of the dimer-dimer interface of *El*STING filament was approximately 1477 Å², like that of *Pc*STING filament (1440 Å²)[21], but reduced to 1088 Å² surface area of *Ri*STING filament. The STING dimer-dimer interface could be divided into four parts: chain A-chain A', chain B-chain B', chain A-chain B', and chain A'-chain B interfaces (Supplementary Fig. 7a, 8a) and described in detail below.

In *Ri*STING filament, the α2 helix and β1-α2 loop from chain A interacted with the α1 helix and α4-α5 loop from chain A', respectively, forming five hydrogen bonds (I180-Q215, Y303-D209, D301-T208, and R300-T208) and numerous hydrophobic interactions (Fig. 5c and Supplementary Fig. 7c, d). In contrast, *Pc*STING filament has one more salt bridge (E306-K220) between chain A and chain A' (Supplementary Fig. 7e). This could be attributed to the unique residue G302 at the end of α4 helix of *Ri*STING, which caused a sharp turn and abolished the α2-α5' interaction but led to closer contact between the β1-α2 loop and α4'-α5' loop (Supplementary Fig. 7d-f). In *El*STING filament, the α2 helix and β1-α2 loop from chain A interacted with the α1 helix and α4-α5 loop from chain A', respectively, forming a salt bridge (D175-R209), two hydrogen bonding pairs (S301-D202 and S301-D204), and extensive hydrophobic contacts (Fig. 5d and Supplementary Fig. 8c-e). The only ionic pair (E175-R209) of *El*STING is also present in *My*STING (E175-K209), but it exists as a hydrogen bond in *Sphingobacterium faecium* STING (*Sf*STING) (E171-N208)[31] and van der Waals interaction in *Ri*STING (Supplementary Fig. 9a).

At the bottom of *Ri*STING filament, the sidechain of N273 located at the α3-α4 loop of chain A forms a H-bond with the sidechain of Y292 of the α4 helix of chain A' (Supplementary Fig. 7g). In contrast, the sidechain of I272 in the α3-α4 loop of chain A from *El*STING forms a continuous hydrophobic interaction with I163, T292, N295, and L296 of chain A', similar to the equivalent residues I272 of *My*STING and V280 of *Sf*STING[31] (Supplementary Fig. 7b). The interactions between chain A and chain B' of *El*STING is mediated through continuous hydrophobic contacts between the α3-α4 loop from both chains, including residues M265, K268, K269, G270, and H271; however, the α3-α4 loop of *Ri*STING dimers contain only N270-N270' van der Waals interactions (Supplementary Fig. 7b, 8b). The interaction between chains A' and B of *Ri*STING involves residues Q181 and L231 of *Ri*STING (Supplementary Fig. 7h), which are absent in *El*STING filament. Moreover, an important ionic pair (E290-R307) in the *Sf*STING dimer-dimer interface[31], which has been validated to block filament formation upon mutagenesis, is completely absent in the *El*STING, *Ri*STING, and *My*STING filaments (Supplementary Fig. 9c). In summary, the detailed dimer-dimer interactions of bacterial STING filaments from five different species have been compared, revealing significant differences and great biodiversity beyond previously reported.

The STING-STING interaction is proved to be the major driving force for *Sf*STING filament formation previously[31]. To validate those observed in *El*STING and *Ri*STING complex structures, we mutated two important interacting residues, D175 and S179, on the dimer-dimer interface of *El*STING to arginine residues, which can disrupt the ionic bond between D175 and R209 as well as cause steric clashes between helices α1 and α2. Gel-filtration analysis demonstrated that cAA could induce the formation of large oligomers of full-length *El*TIR-STING proteins, in contrast to the dimeric assembly in the absence of a ligand (Fig. 5e, f). The molecular weight of the cAA-induced *El*TIR-STING oligomer was 3937−7883 kDa, corresponding to large oligomers (Fig. 5g). The resulting oligomerization mutant *El*TIR-STING^D175R,S179R forms hexamers and dimers in solution regardless of the addition of cAA (Fig. 5h), validating the importance of the interacting residues observed in the *El*STING_cAA complex structure. Conversely, the STING-STING interaction of *Ri*STING filament was validated by phage infection experiments. As shown in Fig. 5i, *E. coli* cells overexpressing full-length *Ri*STING conferred resistance to phage infections at a multiplicity of infection (MOI) of 1 and 0.01, indicating the formation of active *Ri*STING oligomers. In contrast, *E. coli* cells harboring the full-length *Ri*TIR-STING^G302E oligomerization mutant completely lost anti-phage ability similar to that in the control cells, validating the importance of residue G302 for tight oligomer formation in *Ri*STING (Fig. 5i). In addition, the triple mutant *Ri*TIR-STING^N270E,N273E,Y292E, which disrupt the H-bond and van der Waals interactions between STING dimers, also remarkably reduced its anti-phage ability (Fig. 5i). Overall, these data suggest that the STING-STING interactions observed in crystals are important for long filament formation and anti-phage responses.

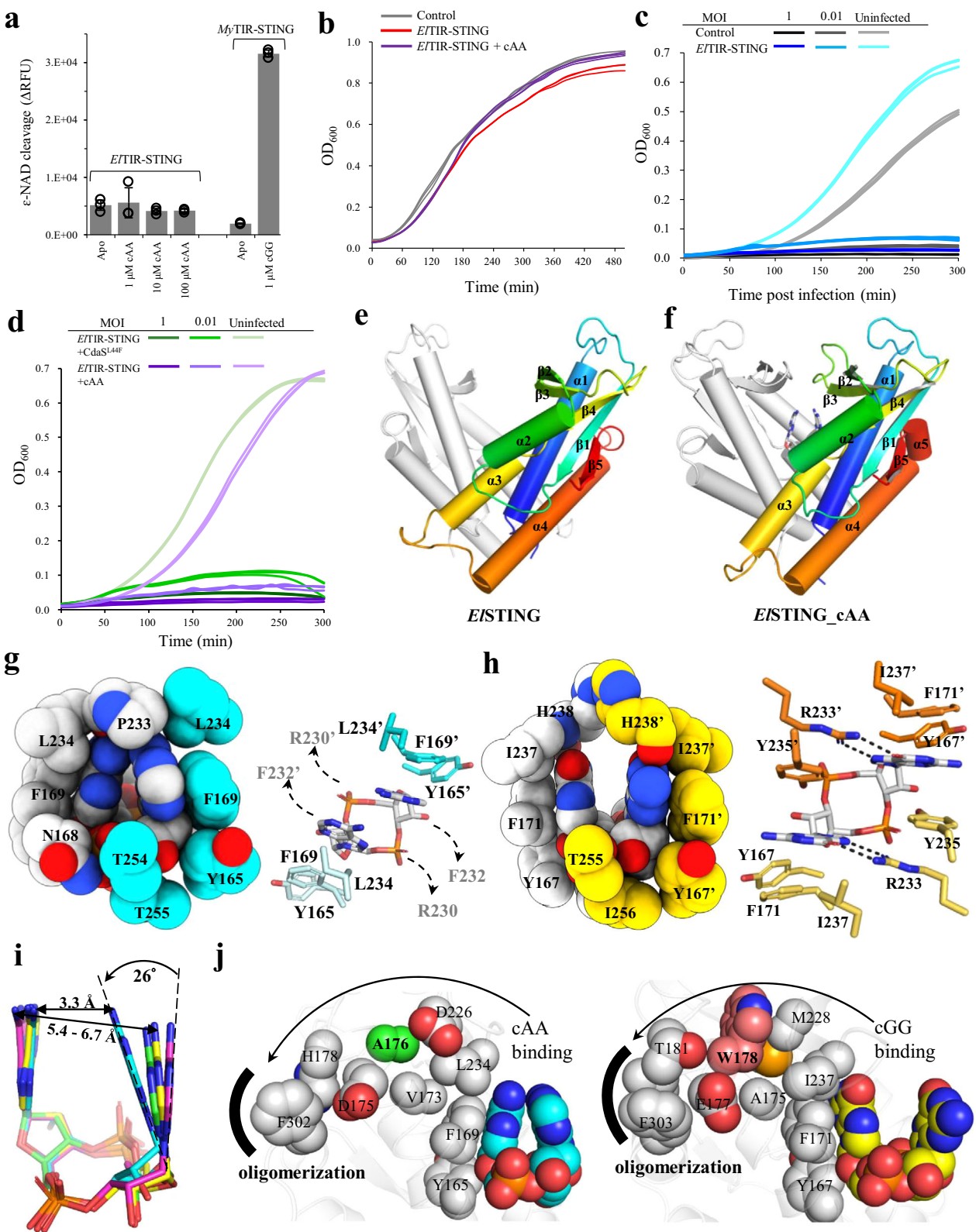

Single-particle electron microscopy (EM) was performed to gain insights into the spiral filament formation by *El*TIR-STING. Interestingly, negative stain EM micrographs showed that *El*TIR-STING oligomerize into two different kinds of filaments: (1) single spiral-shaped protofibrils about 200 nm in length and (2) higher-order fibril filaments extending more than 700 nm (Fig. 5j–l and Supplementary Fig. 10). In comparison with near straightforward *Sf*TIR-STING

filament, *El*TIR-STING proto-fibrils showed larger curvature[31] (Fig. 5j, k). Moreover, it seems that spiral-shaped *El*TIR-STING proto-fibrils undergo large conformational changes to assemble into huge, linear fibril filaments (composed of more than six protofibrils by estimation), whereas *Sf*TIR-STING makes only rigid-body contacts to form double-fibril filaments[31]. Notably, huge *El*TIR-STING fibril filaments could further interact with each other to form a large filament network

**Fig. 4 | Structural and functional characterization of cAA-bound *El*STING.**
**a** NAD+ cleavage activity of *El*TIR-STING. The concentration of the added CDNs (cGG and cAA) is indicated. The NAD+ cleavage activity of *My*TIR-STING in the absence and presence of cGG were served as negative and positive control, respectively. The data were shown as mean ± standard deviation for *n* = 3 independent replicates. **b** Growth curves of *E. coli* cells overexpressing *El*TIR-STING (red) and *E. coli* cells overexpressing *El*TIR-STING with addition of 300 μM cAA (purple) compared with control (empty vector, grey). **c** Growth curves of *E. coli* cells overexpressing *El*TIR-STING (blue) and control cells (empty vector, black) with and without T2 phage infection at a MOI of 1 or 0.01. **d** Growth curves of *E. coli* cells overexpressing both *El*TIR-STING and diadenylate cyclase CdaS^L144F (green) and *E. coli* cells overexpressing *El*TIR-STING with addition of 100 μM cAA (purple) with and without T2 phage infection at a MOI of 1 or 0.01. The experiments in (**b**–**d**) were performed for *n* = 3 biological replicates and each of them is shown. **e, f** Cartoon

representation of V-shaped dimer of (**e**) *El*STING and (**f**) *El*STING_cAA complex. The β-strand lid of *El*STING is partially invisible (residues 226-233), probably due to structurally flexibility in the absence of ligand[3,27]. **g, h** Comparison of CDN recognition mode of (**g**) *El*STING with (**h**) *Pc*STING (PDB: 7EBD). Left, the residues forming hydrophobic contacts surrounding the CDN are shown in spheres. Right, the CDN-interacting residues are shown in sticks for comparison. **i** Structural comparison of the bound cAA between *El*STING (cyan), human STING (green, PDB 6Z15[56]), porcine STING (yellow, PDB 6A03[57]) and sea anemone STING (magenta, PDB 5CFN[27]). The proteins are neglected for simplicity. The distance (Å) between the 2-NH2 group of adenine bases of cAA are indicated. The largest rotational angle between adenine bases of different cAA conformations is ~26°. **j** cAA (left) and cGG (right) induces the conformational changes of *El*STING and *Pc*STING, respectively, leading to exposure of oligomerization interface for filament formation. CDNs and the residues mediated the conformational changes are shown in spheres.

(Fig. 5j), suggesting the existence of distinct oligomerization mechanisms between bacterial STINGs.

## Discussion

In this study, we provide the detailed structural information for antiparallel dimerization of *Ri*STING in the absence of activating signal. The SWAXS results demonstrates that the major form of ligand-free *Ri*STING in solution is anti-parallel dimeric conformation, but the subtle inconsistence between experimental and calculated curves could attribute to the existence of small populations of alternative conformation of *Ri*STING, which is speculated to be ligand-free V-shaped dimer as previously reported *Capnocytophaga granulosa* STING (*Cg*STING)[17]. The V-shaped STING dimer is more ready to bind activating signal than anti-parallel dimer. Once binding of cGG, the equilibrium shifts toward the formation of V-shaped STING dimer with β-strand lid closed, which will further oligomerize into long filaments. Furthermore, conservation analysis using ConSurf webserver[32] demonstrates that most of the residues at the anti-parallel *Ri*STING dimer interface are highly conserved among bacterial STING sequences (Supplementary Fig. 11a). More accurately, up to 10 of 14 residues unique to anti-parallel dimer formation but not V-shaped dimer formation get high conservation scores (≥7, Supplementary Fig. 11b), suggesting that this conformation is evolutionarily conserved across bacteria.

Considering the vast diversity of bacteria living in diverse niches, some of them may encode STING effectors with distinct features. Here we report that dozens of bacterial STINGs contain additional insertion of 14–15 amino acids compared with canonical STINGs and that the insertion sequence correspond to an additional long lid above ligand-binding pocket in *Lr*STING structure. There are two possible explanations for this evolutionary innovation in extremophile STINGs. The first is that to adapt with low-resource environments, the CDN products may be limited, leading to the emergence of additional long lid to maintain longer retention of second messenger in the ligand-binding pocket. Second, we identified a cavity (~2740 Å³) formed between the long lid and original lid, which may bind an unknown small regulatory molecule to fine-tune their anti-phage functions. In summary, our data shed light on the diversity of STING effectors from microorganism in distinct niches.

STING autoinhibition has been a long-standing mystery for researchers in this field. Ergun et al.[4] demonstrated that in the absence of activating signal, the CTT of metazoan STING binds to its oligomerization interface, thus blocking STING polymer formation, which has long been thought to be the key for downstream effector recruitment and activation. However, during the preparation of this manuscript, Liu et al[33]. overturn this view by determining the cryo-EM structure of apo chicken STING polymer that adopt a bilayer with head-to-head interaction. This apo-STING bilayer connect two endoplasmic reticulum membrane together, which prevent from STING trafficking to Golgi and also block TBK1 binding. Interestingly, the long lid of

bacterial STING that autoinhibits itself from activation discovered in *Lr*STING in this study could be considered as a *cis* interaction in contrast to the head-to-head *trans* interaction mediated by lid region of chicken STING.

In this study, we provide two distinct CDN-binding modes of bacterial STING: (a) sandwich-binding and (b) tight clamping. In contrast to the four-layer stack interaction (R233/Y235/Guanine/F171) with cGG in *Pc*STING structure[21], *Ri*STING has two adjacent arginine residues, R233 and R235, which will cause electrostatic repulsion once stacking with each other. To avoid this, *Ri*STING interact with cGG via sandwich-binding mode (R235/Guanine/F171) without additional stack with R233 (Fig. 6). In addition, R233 of *Pc*STING (R230 of *My*STING) forms two hydrogen bonds with both O6 and N7 atoms of the guanine base in contrast to only one H-bond between the N7 atom of the guanine base and R233 of *Ri*STING. Replacement of 6-carbony group (guanine) with 6-NH2 group (adenine) could thus be tolerated by *Ri*STING but make clashes with *Pc*STING, indicating a high preference for guanine base of *Pc*STING/*My*STING than *Ri*STING. Moreover, structural comparison of *El*STING, *Ri*STING, and *Pc*STING/*My*STING further demonstrates that different ligand-binding modes results in different oligomerization interfaces and thus STING filament formation. The rotational packing of *El*STING dimers lead to spiral-shaped filament formation in comparison to the linear STING filaments formed with *Ri*STING and *Pc*STING/*My*STING (Fig. 6). Despite similar linear filament formation, *Ri*STING filament is much more flexible than *Pc*STING/*My*STING filament due to smaller dimerization interface and devoid of electrostatic interactions. In conclusion, we here conducted a more detailed study on bacterial STING and summarized three different forms of CDN-mediated oligomerization mechanisms of bacterial STING, which are highly correlated with their functional performance and might have resulted from their adaptation to different environments under selective pressure.

In this study, we determine a second ligand-free V-shaped bacterial STING, *El*STING, which showed a different open configuration from the currently available *Cg*STING[17]. The ligand-binding pocket of *Cg*STING (~51 Å) is larger than that of *El*STING (~30 Å) (Supplementary Fig. 12a, 12b). CDN binding will induce the closure of the β-strand lids of *Cg*STING and *El*STING in nearly perpendicular directions (Supplementary Fig. 12c, 12d). These structural differences result from the direction of rigid-body rotation upon ligand binding: with one protomer superimposed, the other *El*STING protomer rotated clockwise by ~26°, in contrast to the anti-clockwise rotation of *Cg*STING by ~21° (Supplementary Fig. 12e, 12f). As a result, the ligand-binding pocket of *El*STING retains a nearly identical distance between the two stacking F169 residues; however, the distance between the two stacking residues in *Cg*STING is largely increased, facilitating the direct release of the ligand into the solvent upon inactivation (Supplementary Fig. 12g, 12h). In summary, the two different open configurations of bacterial STINGs may be a consequence of the divergent evolution of bacterial STINGs.

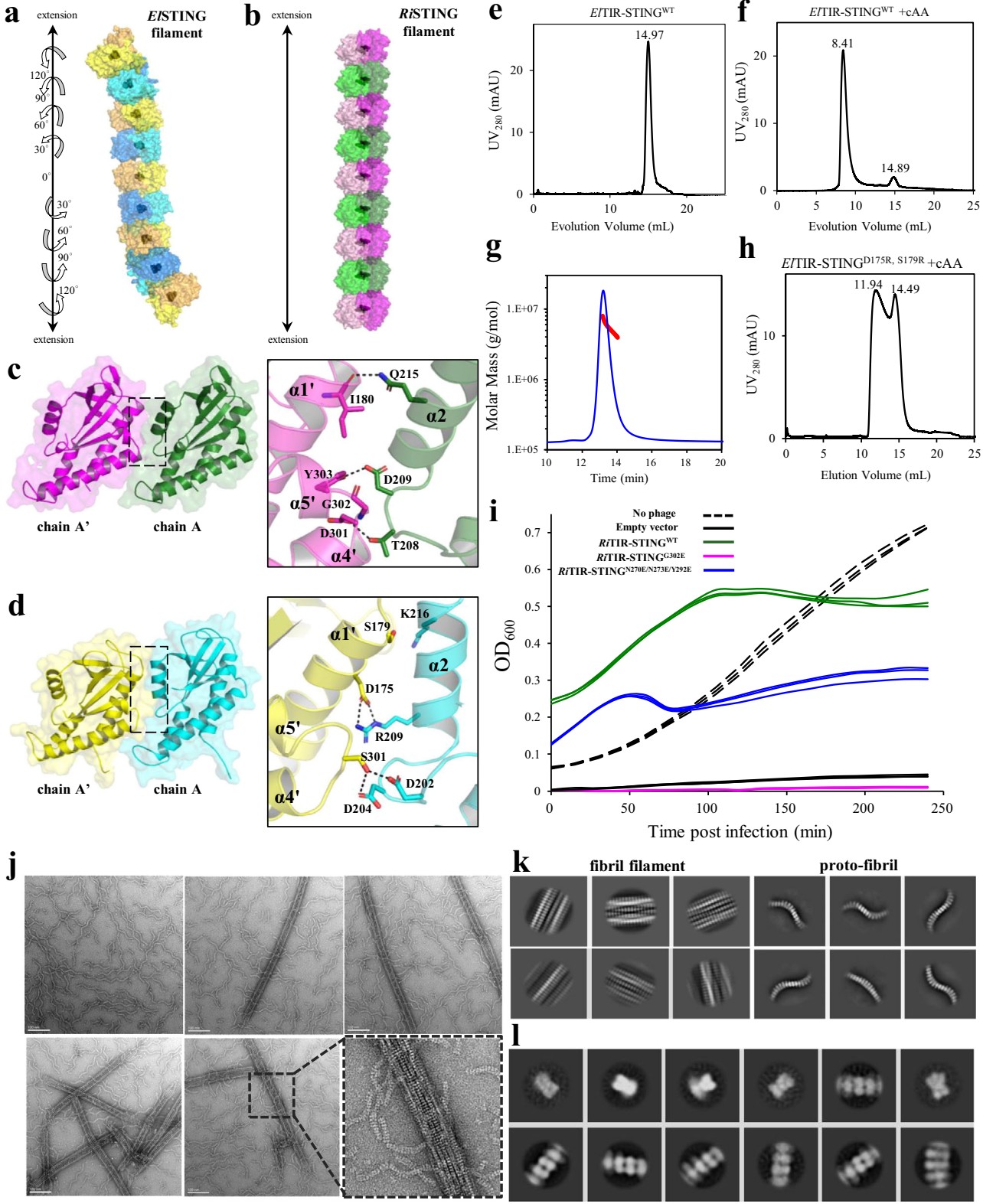

Previous study has summarized the conformational polymorphism of cAA in complex with proteins, which highly correlated with their physiological functions[30]. U-type cAA is the most widely found cAA conformation in complex with receptor proteins, such as DisA[24] and KtrA[34] (Supplementary Fig. 13a). When the glycosidic bonds of two adenine bases are inclined by 45°, the overall V-shaped cAA was named as V-type and was found in receptor DarA[35] and PstA[36] (Supplementary Fig. 13b). The extended (E)-type cAA had a flat conformation with two adenine bases in two opposite directions and have been identified in the immune protein RECON, metabolic enzyme pyruvate carboxylase[37], and the phosphodiesterase GdpP[38] (Supplementary Fig. 13c). The variant of E-type cAA containing one adenine base orthogonal to the other adenine base is referred to as "O"-type cAA (Supplementary Fig. 13d), which binds to the carnitine transporter, OpuC[39]. Here we identified a Δ-type cAA in bacterial STING complex, demonstrating a distinct recognition mode in contrast to U-type and V-type cAA in receptor protein

**Fig. 5 | The structural and functional characterization of *El*STING and *Ri*STING filament. a, b** Extended packing diagrams for (**a**) *El*STING and (**b**) *Ri*STING filament implicated in the complex crystals. The protein models are presented by translucent surface diagram with bound ligand shown in black spheres. The protomers are shown in alternatively different colors. **c, d** Left, the dimer (chain AB)-dimer (chain A'B') interface for the filament formation of (**c**) *Ri*STING and (**d**) *El*STING. The chain B and B' are neglected for clarity. Right, the enlarged view of the interface showing hydrogen-bonding and electrostatic interactions (black dashed lines). **e, f** Gel-filtration analysis of wild-type *El*TIR-STING in the (**e**) absence or (**f**) presence of cAA at 2-fold molar excess. The elution volume of each peak is indicated. The major peak of cAA-bound *El*TIR-STING eluted as large oligomers, while *El*TIR-STING eluted as dimers. **g** SEC-MALS analysis of wild-type *El*TIR-STING in the presence of cAA. The light scattering signal: blue line; the calculated molar mass: red line. **h** Gel-filtration analysis of oligomerization mutant *El*TIR-STING[D175R,S179R] in the presence of 2-fold molar excess of cAA. The double mutant causes electrostatic repulsion

(D175R) and steric clashes (S179R), disrupting the oligomerization interface. **i** Growth curves of *E. coli* cells overexpressing wild-type *Ri*TIR-STING (green), *Ri*TIR-STING[G302E] mutant (magenta), *Ri*TIR-STING[N270E,N273E,Y292E] mutant (blue), and control cells (empty vector, black) with T2 phage infection at a MOI of 0.01. The *E. coli* cells without IPTG induction and T2 phage infection (black dashed lines) serve as negative control. The data from three independent experiments are shown. Changing G302 to glutamate could results in conformational change of α5 helix, steric clashes with β1-α2 loop and electrostatic repulsion with D209, which together abolish the filament formation. **j** Negative-stain electron micrographs of *El*TIR-STING in the presence of equimolar cGG. An enlarged view of the fibrils was shown at its right side. Scale bars, 100 nm. Each image is representative of *n* = 3 micrograph images. **k** 2D class averages showed that *El*TIR-STING form spiral-shaped proto-fibrils and higher-order fibril filaments. The particle dimension is 50 nm. **l** Enlarged view of selected 2D class averages of single proto-fibril of *El*TIR-STING extracted with box size of 15 nm from 99 micrographs.

complex. Given that most gram-positive bacteria and some gram-negative bacteria produce cAA as a second messenger[40], it is likely that *Epilithonimonas lactis* produces it too, using a yet undescribed diadenylate cyclase encoded in its genome. If this is the case, cAA would then activate *El*STING for downstream responses. However, the exact molecular mechanisms underlying cAA-mediated STING activation in *E. lactis* and the physiological relevance of this activation still remain to be studied in depth.

Upon ligand binding, innate immune receptors form large oligomers with different shapes and conformations. The cA3-induced right-handed helical oligomers and cUMP-induced zipper-like interlocking filaments of TIR-domain-containing effectors have been previously reported[41,42]. A notable finding in this study was the direct observation of single, spiral-shaped STING protofibrils, which could further assemble into higher-order fibril filaments and form extensive filament-filament contacts. At the cellular level, the ability to form higher-order assemblies of long filaments covering whole cells is key to rapidly responding to environmental changes or inhibiting phage replication as quickly as possible. Alternatively, the remarkably long STING filament observed in this study ( > 700 nm) is approaching the size of a single bacterium cell (1000–2000 nm long), which could pose steric constraints and create membrane lesions, facilitating cell death and abortive infections. Future characterization of the higher-order fibril filaments of *El*STING is needed to provide a clearer picture of anti-phage STING signaling at the cellular level.

## Methods
### Sample preparation
The gene fragments of full-length or C-terminal STING domain of *Ri*TIR-STING from *Riemerella anatipestifer* Yb2 (GenBank: AKQ40609.1), *El*TIR-STING from *Epilithonimonas lactis* DSM 19921 (NCBI accession: WP_034976296.1), *Lr*TIR-STING from *Larkinella arboricola* DSM 21851 (NCBI accession: WP_111631205.1) and diadenylate cyclase CdaS[L44F] from *Bacillus subtilis* strain 168 (Uniprot accession: O31854) were *E. coli* codon-optimized and synthesized by Genomics BioSci. & Tech. company (Supplementary Table 6). The synthesized genes were subcloned into pET21 (for STING only) or pSol MBP vector (for full-length TIR-STING) to generate C-terminal His6-tagged or N-terminal MBP fusion proteins. Overexpression of the target proteins were induced by addition of 0.5 mM IPTG (for pET21 system) or 0.2% L-rhamnose plus 30 mM nicotinamide (for pSol system), followed by further incubation at 16-20 °C for 20 hours. Two-step protein purification procedure is performed as previously described[21] to achieve high purity and monodispersity. First, target proteins were extracted from cell lysates using Ni-affinity chromatography and eluted by buffer (50 mM Tris-HCl 8.0, 500 mM NaCl, 10% glycerol and 1 mM TCEP) containing imidazole gradient (10, 20, 50, 100, 200 mM). The fractions containing target protein were analyzed by 12 % SDS-PAGE and pooled. The C-terminal His6-tagged STING proteins were directly subject to size-exclusion

chromatography equilibrated with Gel buffer (20 mM Tris-HCl 8.0, 200 mM NaCl, 5% glycerol and 1 mM TCEP) using 16/60 Superdex 200 increase column (Cytiva). The N-terminal MBP-fusion TIR-STING proteins were first processed by TEV protease in buffer containing 25 mM Tris 8.0, 100 mM NaCl, 1 mM DTT, 0.5 mM EDTA, 2% glycerol at 4 °C for 2-3 days. The cleaved MBP and TEV protease were removed by purification using Ni-affinity chromatography again and subsequently subjected to size-exclusion chromatography to achieve purity and monodispersity. The fractions containing purified target proteins were concentrated using Amicon Ultra-15 centrifugal filter unit (Merck Millipore) and store at -80 °C until use.

### Crystallization, data collection and structural determination
The optimal protein concentrations for crystallization were determined by Pre-Crystallization Test (Cat No. HR2-140, Hampton Research). For crystallization of STING_CDN complex, two-fold molar excess of CDN was added into the protein solution. Crystallization conditions screening was performed at 20 °C or 4 °C using sitting-drop vapor-diffusion method. The crystals of ligand-free *Ri*STING were grown in 0.1 M Imidazole/MES pH 6.5, 0.1 M carboxylic acids, 20% v/v ethylene glycol, 10% w/v PEG 8000. The crystals of *Ri*STING_cGG complex were grown in 0.1 M Tris pH 8.5, 0.2 M calcium chloride dihydrate, 40% v/v PEG 4000. The crystals of *El*STING were grown in 0.1 M MOPSO/Bis-Tris pH 7.5, 90 mM lithium/sodium/ potassium sulfate, 15% (w/v) PEG 3000, 20% (v/v) 1, 2, 4-Butanetriol, 1 % (w/v) NDSB 256. The crystals of *El*STING_cAA complex were grown in 0.1 M MES pH 6.0, 0.1 M Sodium chloride, 30 % v/v PEG 3500. The crystals of *Lr*STING_cGG complex were grown in 0.1 M Imidazole pH 7.5, 0.1 M sodium chloride, 20% v/v Jeffamine ED-2001. Before flash-cooling in liquid nitrogen, the protein crystals were cryoprotected with reservoir solution plus 15-25% glycerol or ethylene glycol. The X-ray diffraction data were collected and processed using HKL2000_v722[43] at the National Synchrotron Radiation Research Center (NSRRC) in Hsinchu, Taiwan. The structure of *Ri*STING, *Ri*STING_cGG, *El*STING, *El*STING_cAA and *Lr*STING_cGG were determined by molecular replacement using *Pc*STING_cGG (PDB 7EBD[21]), *My*STING_cGG (PDB 7EBL[21]) or their AlphaFold2 models as searching template, followed by extensive refinement and rebuilt using PHENIX 1.13-2998 and *Coot* 0.9.6[44,45]. All the collected X-ray data and refinement statistics are summarized in Supplementary Table 1, 4, and 5 and the images of a portion of the electron density map for all the reported crystal structures are shown in Supplementary Fig. 14. All the figures containing three-dimensional structures of protein and ligand were depicted using PyMOL 2.3.3[46] and UCSF ChimeraX 1.5[47]. To reconstruct the STING filaments, nine continuous *Ri*STING and *Pc*STING dimers that are symmetry-related by unit-cell translation were generated using PyMOL. Four *El*STING dimers (an octamer unit) was first generated by the same way but further extended to nine continuous dimers via superimposition of *El*STING octamer at the ends[48].

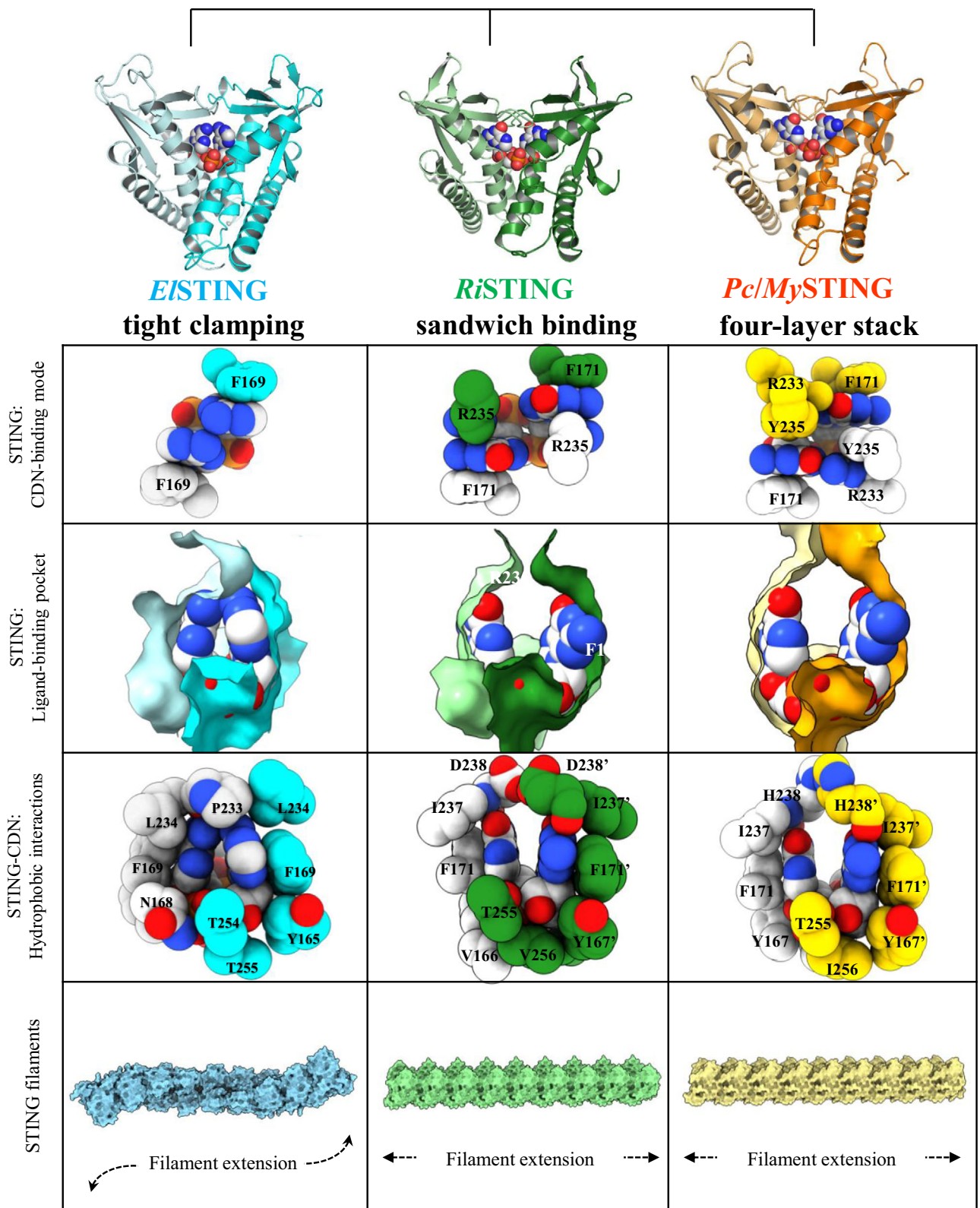

**Fig. 6 | Structure-function evolution of bacterial STINGs.** Structural and functional characterization of *El*STING, *Ri*STING, and previously determined bacterial STINGs reveal differences in CDN recognition and oligomerization, suggesting a model explaining the evolutionary adaption of bacterial STINGs. The different ligand binding modes induced the formation of different STING-STING contacts, thereby the STING filament formation.

## Isothermal titration calorimetry (ITC)

ITC experiments were performed at 25 °C using nano-ITC (TA Instruments). Purified *Ri*STING and *El*STING were exchanged into assay buffer containing 20 mM Tris pH 8.0, 150 mM NaCl. 10 mM stock of cGG (Item No. 17144, Cayman Chemical) and cAA (Item No. 17753, Cayman Chemical) were prepared by dissolving in the same buffer. 20 injections of ligand (0.9-1.2 mM) were sequentially titrated to tested STING protein solution (30-100 µM) with a stirring speed of 300 rpm.

Data were fit to "independent" model to derive ΔH, ΔS, and $K_d$ values using NanoAnalyze v3.12.5 (TA Instruments).

## Dynamic light scattering (DLS)

The DLS experiments were performed using Zetasizer Nano ZS90 (Malvern instruments). Briefly, protein samples containing 1 mg/ml *Ri*STING in the presence or absence of 2-fold molar excess of cGG were prepared and degassed. Polydispersity index (PDI) and globule size were examined at 25 °C.

## Size-exclusion chromatography with multi-angle light scattering (SEC-MALS)

Purified protein samples for SEC-MALS analysis were diluted in buffer containing 50 mM Tris pH 8.0, 200 mM NaCl, 1 mM TCEP to a final concentration of 1 mg/ml. The protein sample was incubated with 0.11 mM cyclic di-nucleotide at 4 °C for 1 hour, followed by centrifugation at 4 °C, 18,000x *g* for 30 mins to remove precipitated proteins. The final protein sample was analyzed using GE AKTA UPC Purifier FPLC system coupled to a Wyatt Optilab T-rex refractive index detector and a Wyatt miniDAWN TREOS to obtain $O.D._{280}$, molar mass and refractive index (dRI).

## Size exclusion chromatography based small- and wide-angle X-ray scattering (SEC-SWAXS)

SEC-SWAXS experiments were performed at the TPS 13 A BioSWAXS beamline of the National Synchrotron Radiation Research Center, Hsinchu, Taiwan[49,50]. The beamline is equipped with two in-vacuum Eiger X 9 M and X 1 M detectors for SWAXS and an in-line HPLC unit (Agilent 1260 series). A Bio SEC-3 silica-based column (pore size 300 Å, Agilent) was equilibrated with a buffer containing cGG before SEC-SWAXS measurements. Sample solutions of 100 μl of *Ri*STING (10 mg/ml), without or with cGG added right before SWAXS measurement (for 2:1 molar ratio of cGG: *Ri*STING dimer), were loaded onto the column with a flow rate of 0.30 ml/min at 10 °C. The eluate from the SEC was directed to the quartz capillary (2 mm dia. and a wall thickness of 20 μm) of the SEC-SWAXS system for X-ray exposure with a 15-keV beam. The SWAXS data were collected continuously with 2 s per frame (with 0.2 s between frames for reducing radiation damage effects) over the elution peak. Buffer scattering was collected before and after the elution peak for background subtraction. The frame data of well-overlapped SWAXS profiles (combined from the two X-ray detectors) were averaged and subtracted with buffer scattering using the TPS 13 A SWAXS Data Reduction Kit (Ver. 3.6)[49]. The data were analyzed respectively using the two programs of all-atom scattering calculation of CRYSOL and *ab* initio reconstruction of protein structure by a chain-like ensemble of dummy residues of GASBOR, that are included in the software package of ATSAS 3.1.3[51]. The presented GASBOR model was averaged from 10 individual runs of the reconstruction.

## NAD$^+$ cleavage activity assay

The NAD$^+$ cleavage activity of the N-terminal TIR domain of full-length TIR-STING proteins were accessed using the previously described protocol[21]. Briefly, 100 μl reaction mixture containing ligand (cGG or cAA) at the indicated concentration and 500 μM ε-NAD (Product No. N2630, Sigma-Aldrich) in the assay buffer (20 mM HEPES pH 7.5, 100 mM KCl) was pre-incubated at room temperature. Tested TIR-STING protein (0.5–1 μM) was then added to start the reaction. The fluorescence signal was continuously monitored for 1 hour at an emission wavelength of 410 nm with excitation at 300 nm using Synergy H1 microplate reader (BioTek). The changes of relative fluorescence units at different time points were recorded.

## Phage amplification and storage

Phage T2 (BCRC 70039) and their host *E. coli* ATCC 11303 (BCRC 13055) were purchased from Bioresource Collection and Research Centre (BCRC), Taiwan. For liquid amplification, the host cells *E. coli* ATCC 11303 were first cultivated in Tryptic soy broth (TSB) at 37 °C for 16–24 hours. 100 μl phage stock was mixed with 300 μl overnight cultured *E. coli* ATCC 11303 and let stand for 15 mins at room temperature. The reaction mixture was then added into 10-ml fresh TSB medium and incubated at 37 °C with 200 rpm shaking for 8-24 hours until the culture became clear. The cultures were centrifuged at 4 °C, 5000x g for 10 mins to remove bacterial host cells and filtration with a 0.22 μm syringe filter. Phage lysate titer was calculated by double agar overlay plaque assay. Phage stocks were stored at 4 °C until use.

## Protein toxicity analysis and phage infection assay in liquid culture of *E. coli* cells

To access the cytotoxicity of TIR-STING proteins and their variants, *E. coli* BL21(DE3) cells which produce endogenous cGG were used. The cells carrying expression plasmid pET21 or pET24 with indicated TIR-STING were first inoculated into 3-ml LB medium and incubated at 37 °C with shaking overnight. The overnight culture was then diluted by fresh LB medium to an optical density at 600 nm ($O.D._{600}$) of ~0.1-0.2, followed by addition of 0.5 mM IPTG and 30 mM nicotinamide to induce the overexpression of tested proteins. The $O.D._{600}$ values were recorded with 15-min interval for 8 hours. To test the effect of cAA on *El*TIR-STING, ectopic addition of 100 μM cAA to culture medium upon induction or co-expression of diadenylate cyclase CdaS$^{L144F}$ from *Bacillus subtilis* strain 168[29] was conducted. To access the anti-phage ability of full-length *Lr*TIR-STING, *E. coli* BL21(DE3) cells containing empty vector, pET24_*Lr*TIR-STING or pET24_*Lr*TIR-STING lid-to-loop mutant was first cultivated at 37 °C overnight and then diluted with fresh medium to $O.D._{600}$ of 0.1-0.2, followed by 0.5 mM IPTG induction and infected with T2 phage at a multiplicity of infection (MOI) of 10 or 0.01. The *E. coli* BL21(DE3) cells without IPTG induction and T2 phage infection was served as control.

## Transmission electron microscopy (TEM)

The protein samples of *El*TIR-STING containing an equimolar ratio cyclic dinucleotide were buffer exchanged with 50 mM Tris pH 8.0, 200 mM NaCl, 1 mM TCEP, and a final concentration of 50 μg mL$^{-1}$. The negative stain grids were prepared as described previously[52,53]. Four microliters of the sample were applied to the carbon film 300 mesh copper grids (Electron Microscope Sciences, PA, USA), which were glow-discharged at 25 mA for 30 s using PELCO easiGlow Glow-discharge Cleaning System (Ted Pella, CA, USA). After 1 min, the excess sample was blotted and the grids were subsequently stained with 4 μL of 2 % uranyl formate (UF, Polysciences, PA, USA) for 1 minute before blotting. The grids were air-dried for a day before image data acquisition. Images were collected under FEI Tecnai G2-F20 Transmission Electron Microscope at 200 kV (Field Electron and Ion Company, OR, USA) with a magnification of 50,000x and a corresponding pixel size of 1.718 Å. A total of 99 micrographs were collected. All the data sets were processed by cryoSPARC v3.3.2[54]. The micrographs were CTF corrected using patch-CTF estimation and the particles were picked by template-based particle picking methodology from all the micrographs with a box size of 500 Å. The datasets were subsequently cleaned with 2D classification to generate 2D class averages containing fibrils or protofibrils.

## Statistics and reproducibility

No statistical method was used to predetermine sample size. No data were excluded from the analyses. ITC experiments shown in Fig. 2b, and Supplementary Fig. 5 were performed in triplicates. DLS experiments were repeated at three times and the representative of them are shown in Fig. 2c. The data from NAD$^+$ cleavage activity assay shown in Fig. 4a represent mean ± standard deviation from three technical replicates. The *E. coli* cytotoxicity assay and phage infection experiments in Figs. 3i, j, 4b, c, d, and 5i were performed

with three biological replicates and each of them is shown. The representative negative-stain micrographs shown in Fig. 5j are selected from 99 collected micrographs in three independent experiments. For refinements of X-ray crystal structures, the 5 % R-free reflection set was randomly determined by PHENIX software. The investigators were not blinded to allocation during experiments and outcome assessment.

## Reporting summary

Further information on research design is available in the Nature Portfolio Reporting Summary linked to this article.

## Data availability

Data are available within the article and supplementary information. The coordinates and structure factors of RiSTING, RiSTING_cGG, ElSTING, ElSTING_cAA, LrSTING_cGG have been deposited in the Protein Data Bank under accession codes 8HYN, 8HY9, 8HY8, 8HWJ, and 8HWI. The protein structures used for analysis in this study are available in the Protein Data Bank under accession codes 4KSY, 5CFN, 6A03, 6Z15, 6WT5, 7EBD, and 7EBL. Protein sequences used in this study are available in the NCBI database under accession code AKQ40609.1 (RiTIR-STING), WP_034976296.1 (ElTIR-STING), and WP_111631205.1 (LrTIR-STING), and Uniprot database under accession code O31854 (CdaS^L44F). Source data are included in the source data file. Source data are provided with this paper.

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

## Acknowledgements

We greatly thank the beamtime allocation by the National Synchrotron Radiation Research Center (NSRRC, Taiwan) and the assistance for the X-ray data collection. We thank the Academia Sinica Cryo-EM Center [AS-CFII-108-110] for data collection, which is funded by the Academia Sinica Core Facility and Innovative Instrument Project. Taiwan Protein Project [AS-KPQ-109-TPP2] is also acknowledged for supporting the cryo-EM facility. This work was supported by grants from the National Science and Technology Council (NSTC) in Taiwan [109-2311-B241-001 and 111-2311-B-039-001-MY3 to Y.C., 110-2113-M-001-050-MY3 and 110-2311-B-001-013-MY3 to S.T.D.H., and 111-2811-M-001-125 to M.K.S.]. This work was also financially supported by Academia Sinica intramural fund, an Academia Sinica Career Development Award, Academia Sinica [AS-CDA-109-L08 to S.T.D.H.] and an Infectious Disease Research Supporting Grant to [AS-IDR-110-08 to S.T.D.H.].

## Author contributions

M.H.H. and Y.C.W. designed and carried out crystallization, structural determination, and all in vitro biochemical and biophysical experiments. C.S.Y. carried out cell toxicity and phage infection assays. K.F.L., J.W.C., O.S., Y.Q.Y. and U.S.J. carried out SWAXS experiments and construct the model. M.K.S., T.W.W., and S.T.D.H. carried out single-particle electron microscopy analysis. Y.C. conceived of the project, wrote the manuscript, and supervised the entire project.

## Competing interests

The authors declare no competing interests.
