## [Peer Review File · Nature Communications]

Structural insights into the regulation, ligand recognition, and oligomerization of bacterial STINGReviewer #1 (Remarks to the Author):

This manuscript by Yang et al reports high-resolution crystal structures and extensive biophysical characterization of several bacterial STING, revealing some interesting diversity in terms of the binding modes of cyclic-dinucleotide ligands and the way in which these STINGs pack together to form oligomers. Some of the highlights include: 1. An anti-parallel dimer of RiSTING, which was proposed to represent an autoinhibited state in the absence of ligand binding; 2. A regulatory role of an extra lid present in STING from some species, which forms an additional layer on top of the canonical lid that embrace the ligand; 3. The "triangular" conformation of cAA bound to EISTING; 4. The different modes of oligomerization of EISTING and RiSTING. These are very interesting findings. In general the paper is well written, and the data presented appear of high quality. I support its publication after the authors address the following minor questions:

1. The experimental data supporting the inhibitory dimer of RiSTING is not definitive. It remains possible that the anti-parallel dimer is a result of crystal packing, or the fact that the construct used for growing crystals was not of the full-length protein. The data from light scattering and SAXS suggest but do not definitely demonstrate the presence of the anti-parallel dimer. I suggest that the authors tone down the claims regarding the inhibitory dimer, or provide mutational analyses of the anti-parallel interface to establish its existence in solution with more confidence.

2. The twisted mode of oligomer of EtSTING as shown in Figure 5a is very interesting. However, the authors need to describe how the oligomer model was generated. If I understand it correctly, the individual dimers in the oligomer are not related by a simple crystallographic translation along one of the unit cell axes. The two dimers in the asymmetric unit together form a tetramer, which pack with a neighboring tetramer to form an octamer. Further extension of the oligomer in the crystal does not exist as the ends are occupied by other symmetry mates that do not belong the oligomer. It is reasonable to reconstruct the oligomer with the octamer unit in the crystal, but it would be better to describe how it is done to avoid confusion.

3. The manuscript, especially the discussion, could be shortened and made more concise. The discussion reiterates many points that are already made clear in the results section.

Xuewu Zhang

Reviewer #2 (Remarks to the Author):

In this manuscript, Yang et al. determined several structures of the LBD domain from bacterial STING with or without CDN bound. The apo-Riemerella anatipestifer STING (RiSTING) adopts a novel "anti-parallel" dimer that is called the auto-inhibited conformation, preventing ligand binding. Bacterial STINGs from extremophiles have an insertion that forms an extra lid lying over the original lid, adding another layer of regulation to STING activation. Furthermore, the authors determined the first bacterial STING/CDA complex in which the CDA could induce STING to form a spiral filament. Interestingly, the protofibril could further form higher-order fibril filaments. In general, this manuscript provides useful information to understand the innate immunity (CBASS system) in bacteria and is of interest to researchers in this field. However, there are also some concerns raised.

Major points:

1, The auto-inhibition of RiSTING is still unclear. The authors did not provide any evidence regarding how this conformation inhibits the activation of the TIR domain. Thus, the anti-parallel conformation of RiSTING could only be recognized as the inactive conformation but not the auto-inhibited conformation. A comparison of the structures of full-length RiSTING with and without ligand could help answer this question.

2, The DLS experiment shows that the size of CDG-bound RiSTING (40.6 nm) is much larger than that of apo-RiSTING (9.0 nm). The authors suggest that this size change is due to the conformational transition from the anti-parallel dimer to the V-shaped dimer. However, the size of

ligand-bound RiSTING is about 5.6 nm based on its structure, which is smaller than the DLS measurement. The reviewer wonders whether the larger size of RiSTING after CDG stimulation is due to the formation of oligomers in the solution, as it is a common property for STING self-association after ligand binding.

3, In Figure 2b, the curve of the integrated heat data in the lower panel appears strange and probably does not fit the raw data in the upper panel.

4, It is not certain whether this anti-parallel conformation is an artifact due to crystal packing, although the authors used SAXS to show its existence in solution. Due to the low resolution of SAXS, the authors may provide additional evidence to support this conclusion. Additionally, it would be interesting to investigate whether this conformation is evolutionarily conserved across bacteria.

5, According to the cell cytotoxicity assay, the LrTIR-STING lid-to-loop variant is a gain-of-function mutation, indicating that the inserted lid plays an important role in autoinhibition rather than attenuating CDG binding.

6, Although CDA can bind to ELSTING and induce STING oligomerization, it cannot activate ELSTING. Additionally, CDA is typically produced by Gram-positive bacteria, while *Epilithonimonas lactis* is a Gram-negative bacterium. Therefore, the CDA/ELSTING complex may not be physiologically relevant.

7, Could the authors provide an explanation for why the CDA/ELSTING oligomer adopts a spiral conformation, in contrast to the canonical CDG/STING oligomer?

Minor points:

1, In line 46 and line 422, "secondary messenger" should be corrected to "second messenger."

2, In line 209, the R199/Q192 interaction is absent, while the R199/D208 interaction is observed in the LrSTING structure.

3, STING autoinhibition has been a long-standing mystery for researchers in this field. Could the authors provide comments on recently published papers (e.g., Ergun et al., Cell 2019 and Liu et al., Mol Cell 2023) in the discussion section?

Point-by-Point Response to the Reviewers' Comments

(Manuscript# NCOMMS-23-22438-T)

REVIEWER COMMENTS

Reviewer #1 (Remarks to the Author):

This manuscript by Yang et al reports high-resolution crystal structures and extensive biophysical characterization of several bacterial STING, revealing some interesting diversity in terms of the binding modes of cyclic-dinucleotide ligands and the way in which these STINGs pack together to form oligomers. Some of the highlights include: 1. An anti-parallel dimer of RiSTING, which was proposed to represent an autoinhibited state in the absence of ligand binding; 2. A regulatory role of an extra lid present in STING from some species, which forms an additional layer on top of the canonical lid that embrace the ligand; 3. The “triangular” conformation of cAA bound to EISTING; 4. The different modes of oligomerization of EISTING and RiSTING. These are very interesting findings. In general the paper is well written, and the data presented appear of high quality. I support its publication after the authors address the following minor questions:

1. The experimental data supporting the inhibitory dimer of RiSTING is not definitive. It remains possible that the anti-parallel dimer is a result of crystal packing, or the fact that the construct used for growing crystals was not of the full-length protein. The data from light scattering and SAXS suggest but do not definitely demonstrate the presence of the anti-parallel dimer. I suggest that the authors tone down the claims regarding the inhibitory dimer, or provide mutational analyses of the anti-parallel interface to establish its existence in solution with more confidence.

2. The twisted mode of oligomer of EtSTING as shown in Figure 5a is very interesting. However, the authors need to describe how the oligomer model was generated. If I understand it correctly, the individual dimers in the oligomer are not related by a simple crystallographic translation along one of the unit cell axes. The two dimers in the asymmetric unit together form a tetramer, which pack with a neighboring tetramer to form an octamer. Further extension of the oligomer in the crystal does not exist as the ends are occupied by other symmetry mates that do not belong the oligomer. It is reasonable to reconstruct the oligomer with the octamer unit in the crystal, but it would be better to describe how it is done to avoid confusion.

3. The manuscript, especially the discussion, could be shortened and made more concise. The discussion reiterates many points that are already made clear in the results section.

Xuewu Zhang

Reviewer #2 (Remarks to the Author):

In this manuscript, Yang et al. determined several structures of the LBD domain from bacterial STING with or without CDN bound. The apo-Riemerella anatipestifer STING (RiSTING) adopts a novel "anti-parallel" dimer that is called the auto-inhibited conformation, preventing ligand binding. Bacterial STINGs from extremophiles have an insertion that forms an extra lid lying over the original lid, adding another layer of regulation to STING activation. Furthermore, the authors determined the first bacterial STING/CDA complex in which the CDA could induce STING to form a spiral filament. Interestingly, the protofiber could further form higher-order fibril filaments. In general, this manuscript provides useful information to understand the innate immunity (CBASS system) in bacteria and is of interest to researchers in this field. However, there are also some concerns raised.

Major points:

1, The auto-inhibition of RiSTING is still unclear. The authors did not provide any evidence regarding how this conformation inhibits the activation of the TIR domain. Thus, the anti-parallel conformation of RiSTING could only be recognized as the inactive conformation but not the auto-inhibited conformation. A comparison of the structures of full-length RiSTING with and without ligand could help answer this question.

2, The DLS experiment shows that the size of CDG-bound RiSTING (40.6 nm) is much larger than that of apo-RiSTING (9.0 nm). The authors suggest that this size change is due to the conformational transition from the anti-parallel dimer to the V-shaped dimer. However, the size of ligand-bound RiSTING is about 5.6 nm based on its structure, which is smaller than the DLS measurement. The reviewer wonders whether the larger size of RiSTING after CDG stimulation is due to the formation of oligomers in the solution, as it is a common property for STING self-association after ligand binding.

3, In Figure 2b, the curve of the integrated heat data in the lower panel appears strange and probably does not fit the raw data in the upper panel.

4, It is not certain whether this anti-parallel conformation is an artifact due to crystal packing, although the authors used SAXS to show its existence in solution. Due to the low resolution of SAXS, the authors may provide additional evidence to support this conclusion. Additionally, it

would be interesting to investigate whether this conformation is evolutionarily conserved across bacteria.

5, According to the cell cytotoxicity assay, the LrTIR-STING lid-to-loop variant is a gain-of-function mutation, indicating that the inserted lid plays an important role in autoinhibition rather than attenuating CDG binding.

6, Although CDA can bind to ELSTING and induce STING oligomerization, it cannot activate ELSTING. Additionally, CDA is typically produced by Gram-positive bacteria, while *Epilithonimonas lactis* is a Gram-negative bacterium. Therefore, the CDA/ELSTING complex may not be physiologically relevant.

7, Could the authors provide an explanation for why the CDA/ELSTING oligomer adopts a spiral conformation, in contrast to the canonical CDG/STING oligomer?

Minor points:

1, In line 46 and line 422, "secondary messenger" should be corrected to "second messenger."

2, In line 209, the R199/Q192 interaction is absent, while the R199/D208 interaction is observed in the LrSTING structure.

3, STING autoinhibition has been a long-standing mystery for researchers in this field. Could the authors provide comments on recently published papers (e.g., Ergun et al., Cell 2019 and Liu et al., Mol Cell 2023) in the discussion section?

Author Rebuttals

Reviewer #1 (Remarks to the Author):

This manuscript by Yang et al reports high-resolution crystal structures and extensive biophysical characterization of several bacterial STING, revealing some interesting diversity in terms of the binding modes of cyclic-dinucleotide ligands and the way in which these STINGs pack together to form oligomers. Some of the highlights include: 1. An anti-parallel dimer of RiSTING, which was proposed to represent an autoinhibited state in the absence of ligand binding; 2. A regulatory role of an extra lid present in STING from some species, which forms an additional layer on top of the canonical lid that embrace the ligand; 3. The “triangular” conformation of cAA bound to ElSTING; 4. The different modes of oligomerization of ElSTING and RiSTING. These are very interesting findings. In general the paper is well written, and the data presented appear of high quality. I support its publication after the authors address the following minor questions:

1. The experimental data supporting the inhibitory dimer of RiSTING is not definitive. It remains possible that the anti-parallel dimer is a result of crystal packing, or the fact that the construct used for growing crystals was not of the full-length protein. The data from light scattering and SAXS suggest but do not definitely demonstrate the presence of the anti-parallel dimer. I suggest that the authors tone down the claims regarding the inhibitory dimer, or provide mutational analyses of the anti-parallel interface to establish its existence in solution with more confidence.

Response: Thanks for reviewer’s comment. We have struggled to provide more evidence by construction of numerous *RiSTING* mutants at the anti-parallel dimer interface including *RiSTING*^{L267A}, *RiSTING*^{L268A}, *RiSTING*^{F272A}, *RiSTING*^{Y279A}, and *RiSTING*^{I282A}. However, some of the mutated *RiSTING* proteins could not be obtained in solubilized form after extensive trials with various conditions including culture mediums, induction temperatures and time, and expression host, while some *RiSTING* variants were unstable and prone to degrade into smaller fragments.

We have tried our best to repeat the SAXS experiments and further improve the resolution of *RiSTING* solution structures by applying wide-angle X-ray scattering (WAXS, together called SWAXS) under a limited grant. The detail method and the results of the combined SWAXS analysis have been newly added to the methods section, lines 543–560 and the main text, lines 179–193, respectively. The figure 2 panel c to f have been replaced with newly updated data (Fig. 2d) and the SWAXS models (Fig. 2e, 2f) and shown as below. The results showed that even in the high-resolution *q*-range data (0.3–0.4 Å⁻¹), the SWAXS data from solution structure of *RiSTING* with added cGG fitted well with the calculated SWAXS curve from the crystal structure of cGG-bound *RiSTING* dimer with small chi square value, indicating that the solution structure is of

similar structural features as that in the crystalline form. In the absence of cGG, most SWAXS data are in good agreement with the crystal structure of anti-parallel *Ri*STING dimer; however, the smeared data in the q -range data near 0.3 \AA^{-1} deviate from the calculated SWAXS profile, suggesting the existence of alternative conformation state of *Ri*STING other than anti-parallel dimeric assembly. These newly updated results provide *Ri*STING solution structures with higher resolution and further confirm the existence of anti-parallel dimer in solution.

Fig. 2. Conformational transition between inactive and active states of bacterial STINGs.

d, CRYSOLOG comparison between the SWAXS data of the solution structures (apo *Ri*STING dimer, red circle; cGG-bound *Ri*STING dimer, black rectangle) and the calculated SWAXS profiles generated by their corresponding crystal structures (solid lines). **e**, **f**, the best-fitted GASBOR models shown in shaded area docked with their corresponding crystal structure of (e) apo *Ri*STING (PDB 8HYN) and (f) *Ri*STING_cGG complex (PDB 8HY9).

To follow the reviewer's valuable comment, we decided to remove the claims regarding the auto-

inhibition of *Ri*STING by anti-parallel dimerization according to the reviewer's suggestion. The modifications are described as below:

- 1) The description in abstract, lines 23–26 “...we determined crystal structure of first auto-inhibited form of bacterial STING, which forms anti-parallel dimer, preventing cyclic dinucleotides (CDN) access. Transition between auto-inhibited form, ligand-free form and ligand-bound form provide on-off switch of bacterial STINGs” has been changed to “...we determined the first crystal structure of anti-parallel dimeric form of bacterial STING, which keep itself in an inactive state by preventing cyclic dinucleotides (CDN) access. Conformational transition between inactive and active states of bacterial STINGs provides an on-off switch for downstream signaling”.
- 2) The first subheading in results section “Bacterial STING is auto-inhibited by anti-parallel dimerization” has been changed to “Bacterial STING adopts a novel anti-parallel dimeric conformation” in line 119.
- 3) The sentence in the end of the second paragraph of results section “... the novel anti-parallel dimeric architecture of ligand-free *Ri*STING is considered as an auto-inhibited state of bacterial STING ...” has been changed to “... the novel anti-parallel dimeric architecture of ligand-free *Ri*STING is considered as an inactive state of bacterial STING ...” in lines 161–163.
- 4) The description in results section “... three different conformation states: (1) anti-parallel auto-inhibited form, (2) V-shaped ligand-free form, and (3) V-shaped ligand-bound form. In the absence of cGG, a dynamic equilibrium probably exists between the anti-parallel auto-inhibited form and the V-shaped ligand-free form of bacterial STINGs. The auto-inhibited form of bacterial STING prevents ligand access ...” has been changed to “...three different conformation states: (1) inactive anti-parallel ligand-free form, (2) inactive V-shaped ligand-free form, and (3) active V-shaped ligand-bound form. In the absence of cGG, bacterial STINGs form both anti-parallel and V-shaped dimeric architectures, which exist in a dynamic equilibrium. The anti-parallel dimerization of bacterial STING prevents ligand access...” in lines 165–168.
- 5) The title of Fig. 2 “Conformational transition between anti-parallel auto-inhibited form, V-shaped ligand-free form, and V-shaped ligand-bound form of bacterial STINGs” has been changed to “Conformational transition between inactive and active states of bacterial STINGs” in line 706, and the legend “... equilibrium between anti-parallel auto-inhibited form and V-shaped ligand-free form” has been changed to “... equilibrium between an anti-parallel dimer and a V-shaped dimer” in line 709.

2. The twisted mode of oligomer of *Et*STING as shown in Figure 5a is very interesting. However, the authors need to describe how the oligomer model was generated. If I understand it correctly, the individual dimers in the oligomer are not related by a simple crystallographic translation along one of the unit cell axes. The two dimers in the asymmetric unit together form a tetramer, which pack with

a neighboring tetramer to form an octamer. Further extension of the oligomer in the crystal does not exist as the ends are occupied by other symmetry mates that do not belong the oligomer. It is reasonable to reconstruct the oligomer with the octamer unit in the crystal, but it would be better to describe how it is done to avoid confusion.

Response: Thanks for reviewer's valuable comment. To avoid confusion, the detail of how to reconstruct STING oligomer has been added to the methods section, lines 514–518 of the revised manuscript and shown as below:

“To reconstruct the STING filaments, nine continuous *Ri*STING and *Pc*STING dimers that are symmetry-related by unit-cell translation were generated using PyMOL. Four *Ei*STING dimers (an octamer unit) was first generated by the same way but further extended to nine continuous dimers via superimposition of *Ei*STING octamer at the ends.”

3. The manuscript, especially the discussion, could be shortened and made more concise. The discussion reiterates many points that are already made clear in the results section.

Xuewu Zhang

Response: Thanks for reviewer's valuable comment. The entire manuscript has been carefully examined and further shortened to make it more concise. The redundant descriptions in the discussion section have been deleted. All the changes in the revised manuscript have been highlighted in red fonts with track changes.

Reviewer #2 (Remarks to the Author):

In this manuscript, Yang et al. determined several structures of the LBD domain from bacterial STING with or without CDN bound. The apo-Riemerella anatipestifer STING (RiSTING) adopts a novel "anti-parallel" dimer that is called the auto-inhibited conformation, preventing ligand binding. Bacterial STINGs from extremophiles have an insertion that forms an extra lid lying over the original lid, adding another layer of regulation to STING activation. Furthermore, the authors determined the first bacterial STING/CDA complex in which the CDA could induce STING to form a spiral filament. Interestingly, the protofiber could further form higher-order fibril filaments. In general, this manuscript provides useful information to understand the innate immunity (CBASS system) in bacteria and is of interest to researchers in this field. However, there are also some concerns raised.

Major points:

1, The auto-inhibition of RiSTING is still unclear. The authors did not provide any evidence regarding how this conformation inhibits the activation of the TIR domain. Thus, the anti-parallel conformation of RiSTING could only be recognized as the inactive conformation but not the auto-inhibited conformation. A comparison of the structures of full-length RiSTING with and without ligand could help answer this question.

Response: Thanks for reviewer's valuable comment. We have replaced the description regarding the anti-parallel conformation of RiSTING with inactive conformation of RiSTING in the revised manuscript. The detailed modifications of the descriptions have been listed as below:

- 1) The description in abstract, lines 23–26 "...we determined crystal structure of first auto-inhibited form of bacterial STING, which forms anti-parallel dimer, preventing cyclic dinucleotides (CDN) access. Transition between auto-inhibited form, ligand-free form and ligand-bound form provide on-off switch of bacterial STINGs" has been changed to "...we determined the first crystal structure of anti-parallel dimeric form of bacterial STING, which keep itself in an inactive state by preventing cyclic dinucleotides (CDN) access. Conformational transition between inactive and active states of bacterial STINGs provides an on-off switch for downstream signaling".**
- 2) The first subheading in results section "Bacterial STING is auto-inhibited by anti-parallel dimerization" has been changed to "Bacterial STING adopts a novel anti-parallel dimeric conformation" in line 119.**
- 3) The sentence in the end of the second paragraph of results section "... the novel anti-parallel dimeric architecture of ligand-free RiSTING is considered as an auto-inhibited state of bacterial STING ..." has been changed to "... the novel anti-parallel dimeric architecture of ligand-free RiSTING is considered as an inactive state of bacterial STING ..." in lines 161–163.**
- 4) The description in results section "... three different conformation states: (1) anti-parallel**

auto-inhibited form, (2) V-shaped ligand-free form, and (3) V-shaped ligand-bound form. In the absence of cGG, a dynamic equilibrium probably exists between the anti-parallel auto-inhibited form and the V-shaped ligand-free form of bacterial STINGs. The auto-inhibited form of bacterial STING prevents ligand access ...” has been changed to “...three different conformation states: (1) inactive anti-parallel ligand-free form, (2) inactive V-shaped ligand-free form, and (3) active V-shaped ligand-bound form. In the absence of cGG, bacterial STINGs form both anti-parallel and V-shaped dimeric architectures, which exist in a dynamic equilibrium. The anti-parallel dimerization of bacterial STING prevents ligand access...” in lines 165–168.

- 5) The title of Fig. 2 “Conformational transition between anti-parallel auto-inhibited form, V-shaped ligand-free form, and V-shaped ligand-bound form of bacterial STINGs” has been changed to “Conformational transition between inactive and active states of bacterial STINGs” in line 706, and the legend “... equilibrium between anti-parallel auto-inhibited form and V-shaped ligand-free form” has been changed to “... equilibrium between an anti-parallel dimer and a V-shaped dimer” in line 709.

To provide atomic structural information of full-length *RiSTING* with and without ligand, we have struggled to determine the crystal structures of them using protein X-ray crystallography. After extensive crystallization trials, we have obtained small, needle-like crystals of full-length *RiSTING* with cGG, which require further optimization to get quality-enough, well-diffracted crystals; however, we failed to obtain any crystals of full-length *RiSTING* without cGG probably due to the high structural flexibility of *RiSTING* in the absence of ligand. To solve this problem, we have collaborated with the research team in Academia Sinica Cryo-EM Facility (ASCEM) to collect cryo-electron microscopy (EM) data of full-length *RiSTING*. Single-particle reconstruction of full-length *RiSTING* structure with and without ligand will be determined in future studies, which will help answer this question.

2, The DLS experiment shows that the size of CDG-bound RiSTING (40.6 nm) is much larger than that of apo-RiSTING (9.0 nm). The authors suggest that this size change is due to the conformational transition from the anti-parallel dimer to the V-shaped dimer. However, the size of ligand-bound RiSTING is about 5.6 nm based on its structure, which is smaller than the DLS measurement. The reviewer wonders whether the larger size of RiSTING after CDG stimulation is due to the formation of oligomers in the solution, as it is a common property for STING self-association after ligand binding.

Response: Yes. The larger size of *RiSTING* after cGG binding is due to the self-association of *RiSTING* in the solution, indicating that *RiSTING* underwent conformational changes from anti-parallel dimer to V-shaped *RiSTING* dimer, which is the building block for *RiSTING* filament formation.

3, In Figure 2b, the curve of the integrated heat data in the lower panel appears strange and probably does not fit the raw data in the upper panel.

Response: Thanks for the reviewer’s valuable comment. The scale of the lower panel has been adjusted so that the integrated heat data match the raw data in the upper panel. The revised Figure 2b has been updated and shown as below.

4, It is not certain whether this anti-parallel conformation is an artifact due to crystal packing, although the authors used SAXS to show its existence in solution. Due to the low resolution of SAXS, the authors may provide additional evidence to support this conclusion. Additionally, it would be interesting to investigate whether this conformation is evolutionarily conserved across bacteria.

Response: Thanks for the reviewer’s valuable comment. To increase the resolution of *RiSTING* solution structures, we have further conducted the wide-angle X-ray scattering (WAXS), which can provide more accurate structural information at the atomic scale. The detail method and the results of combined SAXS and WAXS analysis (together called SWAXS) have been newly added to the methods section, lines 543–560 and the main text, lines 179–193, respectively. As shown in Figure 2d, most of the SWAXS data of the apo *RiSTING* in solution fitted well with the calculated SWAXS profile from the crystal structure of anti-parallel *RiSTING* dimer determined in this study. In addition, the reconstructed GASBOR model from the SWAXS data of apo *RiSTING* in solution superimposes well with its crystalline form (Figure 2e).

The evolutionary conservation of the residues involved in anti-parallel dimer formation of *RiSTING* has been newly added to Supplementary Fig. S10 and shown as below. The description of the results of this conservation analysis was newly added to the discussion section, lines 364–369 and shown as below:

“Furthermore, conservation analysis using ConSurf webserver³² demonstrated that most of the residues at the anti-parallel *RiSTING* dimer interface are highly conserved among bacterial STING sequences (Supplementary Fig. 10a). More accurately, up to 10 of 14 residues unique to anti-parallel dimer formation but not V-shaped dimer formation get high conservation scores (\geq

7, Supplementary Fig. 10b), suggesting that this conformation is evolutionarily conserved across bacteria.”

Supplementary Figure 10. Conservation analysis of residues involved in anti-parallel dimer formation of apo *Ri*STING using ConSurf webserver.

(a) The residues of one of the two protomers of anti-parallel *Ri*STING dimer are colored according to the calculated conservation scores. Conservation scores range from 1 to 9 with increasing conservation are indicated. (b) The semi-transparent cartoon model of *Ri*STING protomer colored according to conservation scores. The residues E169, F228, R235, L267, L268, D271, Y279, I282, L283, and E286 of *Ri*STING involved only in anti-parallel dimer formation, but not V-shaped dimer formation are shown in sticks and labeled.

5, According to the cell cytotoxicity assay, the *Lr*TIR-STING lid-to-loop variant is a gain-of-function mutation, indicating that the inserted lid plays an important role in autoinhibition rather than attenuating CDG binding.

Response: Thanks for reviewer’s valuable comment. The two sentences “... these data suggested the additional long lid of *Lr*STING probably affects the CDN access to ligand-binding pocket by

posing physical hindrance, leading to ...” and “Mutating the lid to random coil facilitates the CDN access and binding, elevating the cell toxicity and anti-phage ability of *LrSTING* ...” have been changed to “...these data suggested that formation of additional long lid of *LrSTING* results in autoinhibition, leading to ...” and “Mutating the lid to random coil relieve the autoinhibition, thereby elevating the cell toxicity and anti-phage ability of *LrSTING* ... ”, respectively, in main text, lines 226–230.

6, Although CDA can bind to ELSTING and induce STING oligomerization, it cannot activate ELSTING. Additionally, CDA is typically produced by Gram-positive bacteria, while Epilithonimonas lactis is a Gram-negative bacterium. Therefore, the CDA/ELSTING complex may not be physiologically relevant.

Response: Although the diadenylyl cyclase enzymes were most identified in gram-positive bacteria, some gram-negative bacteria, including members in phyla Bacteroidetes, Cyanobacteria, and Chlamydiae, have been found to produce CDA (Corrigan et al., *Nat Rev Microbiol.* 2013; 11:513-524.). In this study, we provide solid evidence that *EISTING* specifically binds CDA via isothermal titration calorimetry (ITC) and tightly clamp CDA in *EISTING* CDN-binding pocket revealed by X-ray crystallography. Extensive biochemical and physiological studies will be conducted in the future to understand whether *Epilithonimonas lactis* produced CDA and what the immune responses mediated by CDA/*EISTING* complex.

7, Could the authors provide an explanation for why the CDA/ELSTING oligomer adopts a spiral conformation, in contrast to the canonical CDG/STING oligomer?

Response: Thanks for reviewer’s valuable comment. The more compact binding of CDA in the ligand-binding pocket of *EISTING* and unique ion pair interaction (D175-R209) on the *EISTING* dimer-dimer interface leads to the formation of spiral filaments with a dimer rotation of $\sim 30^\circ$ along the oligomerization axis. In contrast, CDG binding induce the formation of V-shaped STING dimer that is slightly larger than that of CDA-bound *EISTING*, thus exposing different oligomerization interface with distinct hydrophilic and hydrophobic interactions, resulting in the formation of linear filament without dimer rotation. The detailed structural comparison and the rationales for STING filament formation of both CDA/*EISTING* oligomer and canonical CDG/STING oligomer have been described in results section, lines 285–322 and discussion section, lines 391–410.

Minor points:

1, In line 46 and line 422, "secondary messenger" should be corrected to "second messenger."

Response: Thanks for reviewer’s comment. This error has been corrected in line 46 of revised

manuscript. The sentence originally in line 422 has been deleted.

2, In line 209, the R199/Q192 interaction is absent, while the R199/D208 interaction is observed in the *LrSTING* structure.

Response: Thanks for reviewer's comment. The sentence "...form two interchain H-bonds (K202/T206 and R199/Q192)" has been changed to "... form one interchain H-bond (K202/T206) and one salt-bridge (R199/D208)" in line 215 of the revised manuscript. Figure 3g has been replaced with new one with correct interactions and shown as below.

3, *STING* autoinhibition has been a long-standing mystery for researchers in this field. Could the authors provide comments on recently published papers (e.g., Ergun et al., *Cell* 2019 and Liu et al., *Mol Cell* 2023) in the discussion section?

Response: Thanks for reviewer's suggestions. We have added the comments on these two papers in the discussion section, lines 381–390 and shown as below:

“*STING* autoinhibition has been a long-standing mystery for researchers in this field. Ergun et al.⁴ first demonstrated that in the absence of activating signal, the CTT of metazoan *STING* binds to its oligomerization interface, thus blocking *STING* polymer formation, which has long been thought to be the key for downstream effector recruitment and activation. However, during the preparation of this manuscript, Liu et al.³³ overturn this view by determining the cryo-EM structure of apo chicken *STING* polymer that adopt a bilayer with head-to-head interaction. This apo-*STING* bilayer connect two endoplasmic reticulum membrane together, which prevent from *STING* trafficking to Golgi and also block TBK1 binding. Interestingly, the long lid of bacterial *STING* that autoinhibits itself from activation discovered in *LrSTING* in this study could be considered as a *cis* interaction in contrast to the head-to-head *trans* interaction mediated by lid region of chicken *STING*.”

Reviewer #1 (Remarks to the Author):

The authors have addressed my previous concerns. I have no further questions.

Reviewer #2 (Remarks to the Author):

In the revised manuscript, the authors conducted additional experiments to bolster the solidity of their conclusions. For example, the new SWAXS data demonstrates the existence of the novel "anti-parallel" dimer in the solution. However, the physiological aspect of the CDA/STING complex remains weak and may mislead readers. The reviewer suggests that the authors may improve it by rewriting the discussion section to make it clearer before publication.

Point-by-Point Response to the Reviewers' Comments

(Manuscript# NCOMMS-23-22438A)

Author Rebuttals

Reviewer #1 (Remarks to the Author):

The authors have addressed my previous concerns. I have no further questions.

Response: We greatly appreciate the reviewer for the invaluable and insightful comments and suggestions to make our manuscript better.

Reviewer #2 (Remarks to the Author):

In the revised manuscript, the authors conducted additional experiments to bolster the solidity of their conclusions. For example, the new SWAXS data demonstrates the existence of the novel "anti-parallel" dimer in the solution. However, the physiological aspect of the CDA/STING complex remains weak and may mislead readers. The reviewer suggests that the authors may improve it by rewriting the discussion section to make it clearer before publication.

Response: Thanks for the reviewer's comments and suggestions. To clarify this issue, the additional comments "Given that second messenger cAA is produced by most gram-positive bacteria and some gram-negative bacteria, including members in phyla Bacteroidetes, Cyanobacteria, and Chlamydiae, it is likely that *Epilithonimonas lactis* produced cAA by unknown diadenylate cyclase encoded in its genome and activates *EISTING* for downstream responses. The exact molecular mechanisms underlying cAA-mediated STING activation and the physiological outcomes will need further investigation in the future" have been added in the discussion section, lines 509–514 of the revised manuscript.